# Psilocin fosters neuroplasticity in iPSC-derived human cortical neurons

Malin Schmidt[1,2,3]*, Anne Hoffrichter[1,2,3], Mahnaz Davoudi[1,2,3], Sandra Horschitz[1,2,3], Thorsten Lau[1,2,3,4], Marcus W Meinhardt[5,6], Rainer Spanagel[5,7], Julia Ladewig[1,2,3], Georg Köhr[8], Philipp Koch[1,2,3,7]*

[1]Department of Translational Brain Research, Central Institute of Mental Health (ZI), University of Heidelberg/ Medical Faculty Mannheim, Mannheim, Germany; [2]Hector Institute for Translational Brain Research (HITBR gGmbH), Mannheim, Germany; [3]German Cancer Research Center (DKFZ), Heidelberg, Germany; [4]Department of Neuroanatomy, Mannheim Centre for Translational Neuroscience (MCTN), Medical Faculty Mannheim, Heidelberg University, Mannheim, Germany; [5]Institute for Psychopharmacology, Central Institute of Mental Health (ZI), University of Heidelberg/Medical Faculty Mannheim, Mannheim, Germany; [6]Molecular Neuroimaging, Central Institute of Mental Health (ZI), University of Heidelberg/ Medical Faculty Mannheim, Mannheim, Germany; [7]German Center for Mental Health (DZPG), partner site Mannheim-Heidelberg-Ulm, Mannheim, Germany; [8]Department of Neurophysiology, Mannheim Center for Translational Neuroscience (MCTN), University of Heidelberg/Medical Faculty Mannheim, Mannheim, Germany

*For correspondence:
Malin.Schmidt@zi-mannheim.de (MS);
philipp.koch@zi-mannheim.de (PK)

Competing interest: The authors declare that no competing interests exist.

## eLife Assessment

This **fundamental** study reports the effects of the psychedelic drug psilocin on iPSC-derived human cortical neurons, analyzing different aspects of structural and functional neuronal plasticity. The evidence is **convincing** and supports the value of using iPSC-derived human cortical neurons for testing the potentially translational effects of psilocin and other psychedelic-related compounds.

**Abstract** Psilocybin is studied as innovative medication in anxiety, substance abuse and treatment-resistant depression. Animal studies show that psychedelics promote neuronal plasticity by strengthening synaptic responses and protein synthesis. However, the exact molecular and cellular changes induced by psilocybin in the human brain are not known. Here, we treated human cortical neurons derived from induced pluripotent stem cells with the 5-HT2A receptor agonist psilocin – the psychoactive metabolite of psilocybin. We analyzed how exposure to psilocin affects gene expression, neuronal morphology, synaptic markers and neuronal function. Psilocin provoked a 5-HT2A-R-mediated augmentation of BDNF abundance. Transcriptomic profiling identified gene expression signatures priming neurons to neuroplasticity. On a morphological level, psilocin induced enhanced neuronal complexity and increased expression of synaptic proteins, in particular in the postsynaptic compartment. Consistently, we observed an increased excitability and enhanced synaptic network activity in neurons treated with psilocin. In conclusion, exposure of human neurons to psilocin might induce a state of enhanced neuronal plasticity, which could explain why psilocin is beneficial in the treatment of neuropsychiatric disorders where synaptic dysfunctions are discussed.

## Introduction

Since the 1990s, the mind-altering psychedelics have become again the subject of an intriguing stream of research in psychiatric disorders (*Nichols et al., 2017*; *Carhart-Harris and Goodwin, 2017*). Particularly, in times where the development and the approval of new medications is decreasing and the number of patients suffering from psychiatric disorders is rising, there is a huge need for new therapeutic interventions (*Miller, 2010*; *Schenberg, 2018*). Compared to classical drugs, the broad therapeutic effect of psychedelics is rapid, robust and can be long-lasting even after single administration (*Vargas et al., 2021*). In particular, the 5-hydroxytryptamine receptor 2A (5-HT2A-R) targeting psilocybin, the compound of the so-called 'magic mushrooms', is discussed as a fast-acting and long-lasting antidepressant in treatment-resistant depression (TRD), anxiety, obsessive-compulsive disorder and addiction (*Carhart-Harris et al., 2018*; *Grob et al., 2011*; *Ross et al., 2016*; *Moreno et al., 2006*; *Bogenschutz et al., 2022*; *Johnson et al., 2017*). Not surprisingly, psilocybin is stated as 'breakthrough therapy' by the United States Food and Drug Administration (FDA) in the treatment of depression since 2019. While we know that 5-HT2A-R activation plays a principal role in serotonergic psychedelic-mediated behavioral and cellular response (*Holze et al., 2021*; *Nichols, 2016*; *Rickli et al., 2016*; *Glennon et al., 1984*; *Vollenweider et al., 1998*; *González-Maeso et al., 2007*; *Kometer et al., 2013*; *Kraehenmann et al., 2017*; *Preller et al., 2017*; *Vollenweider et al., 1997*; *Marek, 2018*), the molecular and cellular changes induced in the brain – on a single cell and network level are barely understood. The recent demonstration of intracellular 5-HT2A-Rs additionally increased the complexity of psychedelic actions (*Vargas et al., 2023*). The administration of psychedelics may enable brain network resetting (*Nichols et al., 2017*) by generating a plastic cellular state in which synaptic remodeling and augmentation of neuroplasticity-associated proteins and genes are likely (*de Vos et al., 2021*). Indeed, biological evidence for the 'resetting' hypothesis comes from a pioneering study by Olson and colleagues in 2018 which showed that the treatment with serotonergic psychedelics increases the synthesis of synaptic proteins, strengthens synaptic responses, and fosters neurito- and synaptogenesis in rat cortical neurons (*Ly et al., 2018*). The group therefore introduced the term 'psychoplastogen' (Greek: psych- [mind], -plast [molded], -gen [producing]) for underlining their plasticity-promoting properties. Moreover, serotonergic psychedelics have already been shown to promote the 5-HT2A-R-mediated growth of dendritic spines and modulate neurotransmission (*Ly et al., 2018*; *Shao et al., 2021*; *Dakic et al., 2017*). As molecular and cellular psychedelic research is recently based nearly exclusively on animal studies, the question emerged whether those insights can be translated to the human brain. Psychiatric disorders and psychedelic effects are complex, multi-symptomatic, and therefore often difficult to study in non-human model organisms. Most importantly, drugs that are efficiently tested in psychiatric animal models might not be necessarily transferable to the human system (*Howe et al., 2018*). In that context, induced human pluripotent stem cells (iPSCs) have emerged as a powerful tool to generate neurons and neuronal circuitries, model brain disorders and identify the molecular mechanisms of drug interventions (*Zahumenska et al., 2020*).

Here, we explored the molecular, transcriptional, morphometric and functional consequences of the psychoactive 5-HT2A-R agonist psilocin, the active metabolite of psilocybin in human iPSC-derived cortical neurons. We demonstrate that psilocin leads to a set of molecular, morphological and functional changes that start shortly after administration and manifest in time. These include an increase in brain-derived neurotrophic factor (BDNF) expression and an activation of gene expression programs associated with neuromodulation and plasticity, resulting in increased neuronal complexity, synaptogenesis and changes in neuronal network function. Thus, our study provides the first evidence that the 5-HT2A-R agonist psilocybin activates widespread neuroplastic programs in human neurons.

## Results

### Differentiation of human iPSCs into human cortical neurons

As an experimental model to study the effects of psilocin in human neurons and neuronal networks, we differentiated human iPSCs (expressing the pluripotency-associated transcription factors OCT4 and SOX2) into neural progenitor cells, which we further differentiated into neurons of mainly dorsal forebrain identity (*Figure 1A*, *Figure 1—figure supplement 1A and B*; *Chambers et al., 2009*; *Kemp et al., 2016*; *Telezhkin et al., 2016*; *Sen et al., 2016*). Neural progenitor cells express typical neural stem/progenitor cell markers such as NESTIN or SOX2 as well as the dorsal forebrain-associated

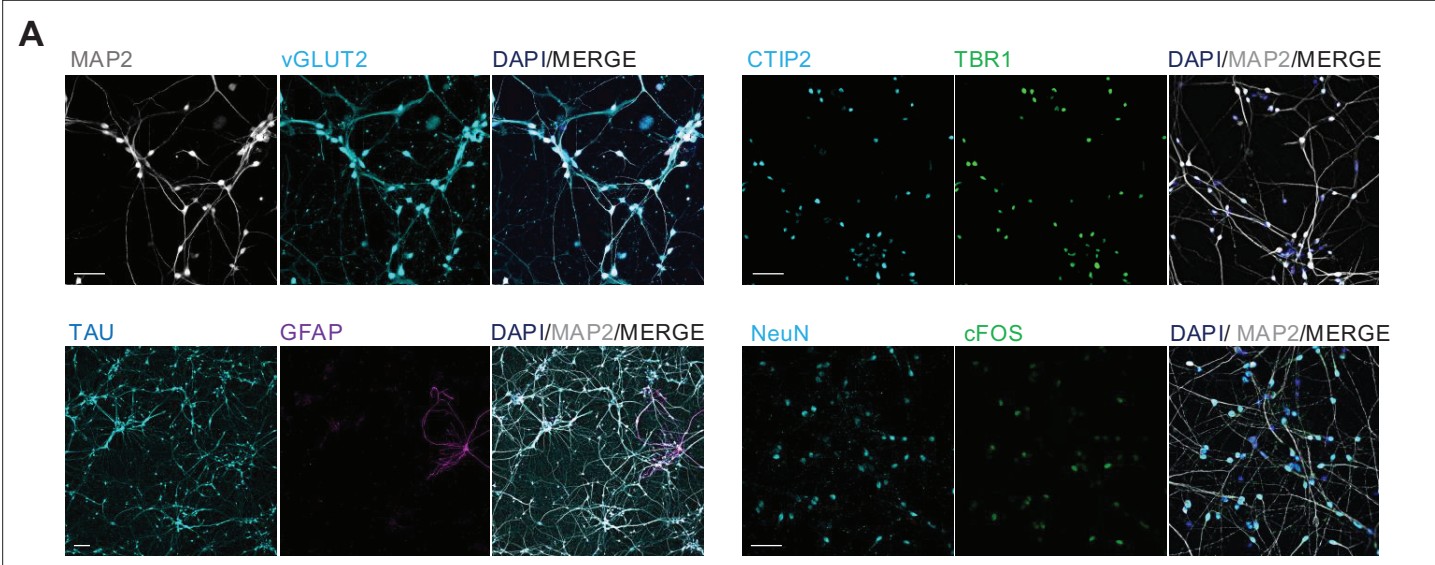

**Figure 1.** Validation of mature cortical neuron properties. (**A**) hiPSCs were differentiated *in vitro* into glutamatergic cortical neurons over a neuronal progenitor step since the cerebral cortex is a key region for psychedelic effects and psychiatric disorders. Mature 40-day-old cortical neurons were glutamatergic (vGLUT2 expression) and expressed cortical layer markers like TBR1, CTIP2 and the neuronal markers NeuN, MAP2, and TAU. cFOS expression is a sign for neuronal activity. GFAP expression indicated a low amount of astrocytes. Scale bar: 50 μm, for GFAP staining 100 μm.

The online version of this article includes the following source data and figure supplement(s) for figure 1:

**Figure supplement 1.** Validation of iPSC and neural progenitor properties and *HTR2A* gene expression.

**Figure supplement 1—source data 1.** RT-PCR analysis revealed expression of neuronal subtype markers, neurotransmitter receptors, neuronal activity markers, neuronal markers and synapse-associated genes in cortical neurons (uncropped, labeled).

**Figure supplement 1—source data 2.** RT-PCR analysis revealed expression of neuronal subtype markers, neurotransmitter receptors, neuronal activity markers, neuronal markers and synapse-associated genes in cortical neurons (uncropped).

**Figure supplement 1—source data 3.** RT-PCR of the HTR2A gene (expressed in commercialized samples of fetal and adult brain) confirmed expression of 5-HT2A-R in cortical neurons (uncropped, labeled).

**Figure supplement 1—source data 4.** RT-PCR of the HTR2A gene (also expressed in commercialized samples of fetal and adult brain) confirmed expression of 5-HT2A-R in cortical neurons (uncropped).

transcription factors PAX6 and are negative for FOXA2, as floorplate/midbrain marker (*Figure 1—figure supplement 1B*). Neurons differentiated from these progenitors for >5 weeks express the neuronal antigen NeuN, the dendritic marker MAP2, and the axonal marker TAU. Most neurons express TBR1 and/or CTIP2, transcription factors typically found in Layer 5/6 neurons of the human cortex (*Figure 1A*). Neuronal activity in mature neurons is indicated by expression of the activity-dependent immediate early gene cFOS (*Figure 1A*). Astrocytes expressing glial fibrillary acid protein (GFAP) were found occasionally (*Figure 1A*). These represent the only contamination in the otherwise purely neuronal culture, which contains no residual progenitors (*Figure 1—figure supplement 1C*). 5-HT2A-R expression was confirmed by PCR (*Figure 1—figure supplement 1D*; *Colaço et al., 2020*; *de Almeida et al., 2019*).

## Psilocin induces BDNF expression and downstream signaling in human neurons

Psychedelics have been shown to efficiently increase levels of neurotrophic factors, such as BDNF (*Holze et al., 2021*; *Ly et al., 2018*; *Colaço et al., 2020*; *de Almeida et al., 2019*). We thus addressed BDNF expression in human neurons following exposure to psilocin. We analyzed the effect of different psilocin concentrations (ranging from 10 nM to 10 μM) exposed for 10 min and also at 24 hrs following exposure. We observed a dosage-dependent increase in the abundance of BDNF-positive particles (quantified as particles per μm neurite length) which reached significance at a concentration of 10 μM psilocin (*Figure 2—figure supplement 1A and B*). A longer treatment (6 hrs instead of 10 min) with a lower concentration (100 nM instead of 10 μM) of psilocin also induced BDNF abundance significantly,

but less strikingly (*Figure 2—figure supplement 1C–D*). We thus continued to use 10 µM psilocin in the following experiments. The clear induction of BDNF particle density was reproduced in four biologically independent control cell lines and several differentiation batches for each line (from MCtrl = 1.5 ± 1.2; MPsi = 2.6 ± 1.9; *Figure 2A–C*). When comparing BDNF densities independently in dendrites (MAP2-positive) and axons (Tau-positive), the significant increase in BDNF could be assigned to both neuronal compartments to a comparable extent (*Figure 2—figure supplement 1E and F*). Also, at 48 hrs following the exposure to psilocin, a significant increase in BDNF particles was observed compared to the untreated condition (*Figure 2—figure supplement 1G*). The increase in BDNF abundance was 5-HT2A-R-mediated as treatment of the neurons with the specific 5-HT2A-R antagonist ketanserin prevented BDNF upregulation by psilocin (*Figure 2D and E*; from MCtrl = 1.8 ± 0.8; MPsi = 3.4 ± 1.7; MPsi + Ket = 1.5±0.7; MKet = 1.4 ± 1.2). Upregulation of BDNF could also be prevented by blocking CME with dynasore or protein kinase C (PKC) with the PKC inhibitor chelerythrine, indicating that CME and PKC activation are critically involved in this process (*Figure 2F and G*; from MCtrl = 3.1 ± 1.5 to MPsi = 4.6 ± 2.4; MPsi +Chel = 3±1.7; MPsi + D = 3.4±1). Upregulation of BDNF was also validated by Western immunoblot where we observed an increase in the levels of pro-BDNF and were able to detect mature BDNF at approximately 14 kDa, which, under baseline conditions, was below the detection level (*Figure 2—figure supplement 1H*, *Figure 2—figure supplement 1H*). Increased BDNF signaling in psilocin-treated neurons is indicated by an increase in the phosphorylation of the BDNF receptor TrkB (*Figure 2—figure supplement 1H*, *Figure 2—figure supplement 1H*). Furthermore, as a downstream target of BDNF signaling, we observed an increase in the phosphorylation of AKT at Ser 473 1 day (*Figure 2—figure supplement 1I*, *Figure 2—figure supplement 1I*) and 3 days after psilocin exposure (*Figure 2H and I*, *Figure 2H*), which was reversed by ketanserin treatment (from MCtrl = 0.8 ± 0.2; MPsi = 1.3 ± 0.2; MKet = 0.9 ± 0.2).

## Psilocin induces gene expression changes associated with axonal and synaptic plasticity

To analyze the influence of psilocin on global gene expression in cortical neurons, we performed whole transcriptome sequencing 1 day and 3 days following a single 10 min administration with 10 µM psilocin in neurons from two independent genetic backgrounds. GO enrichment and KEGG pathway analysis comparing cells 1 day and 3 days after psilocin administration with the respective controls revealed an enrichment of significantly affected genes in many ontologies and pathways associated with axonal growth and synaptic remodeling, plasticity and learning, memory and cognition (*Figure 3A*, *Figure 3—figure supplement 1A*). Most significant changes occurred within the first 24 hrs which is reflected when plotting the significantly changed genes only for selected GO term enriched at both time points (*Figure 3B*, *Figure 3—figure supplement 1B*). When looking at a more global gene expression level of all genes included in the respective GOs, gene expression of GOs associated with axonal outgrowth shows a generally higher expression at day 1, which is further increased at day 3. In contrast, genes associated with synaptic organization, plasticity and learning, memory and cognition show a trend towards decreased expression at day 1 and a generally stronger increase at day 3 (*Figure 3C*, *Figure 3—figure supplement 1C*). The overlapping assignment of significant differentially expressed genes to the aforementioned GOs underlines their importance in long-term plasticity (e.g., *CDK5*, *CAMK*), synapse formation (e.g., *SYN1*) and axonal and neurite growth (e.g., *GAP43*) and neuronal structure marker (e.g., *MAPT*; *Figure 3D*). The latter is consistent with increasing staining intensity for TAU (*Figure 2*). We further found a strongly significant upregulation of immediate early genes (IEGs; e.g., *FOSL2*, *JUND*, *EGR1*, *ARC*, *FOSB*) and of the excitatory glutamatergic AMPA/NMDA receptor genes (*GRIA/GRIN*) after psilocin administration (*Figure 3E*). 1 day but not 3 days of psilocin treatment more strongly upregulated the AMPA genes *GRIA1, 2, 3* than the NMDA genes *GRIN1, 2B, 2C* (*Figure 3E*), supporting modulation of glutamatergic excitatory signalling genes upon psilocin treatment. These effects could be reversed upon co-treatment with the antagonist ketanserin, indicating that most changes are 5-HT2A-R mediated (*Figure 3F*).

## Psilocin increases neurite complexity

The gene expression analysis indicated an induction of morphometric changes by psilocin, in particular on neurites. Therefore, we assessed neuronal complexity by performing Sholl analysis in neurons differentiated for >40 days at 24 hrs and 48 hrs following psilocin exposure. To identify the dendritic

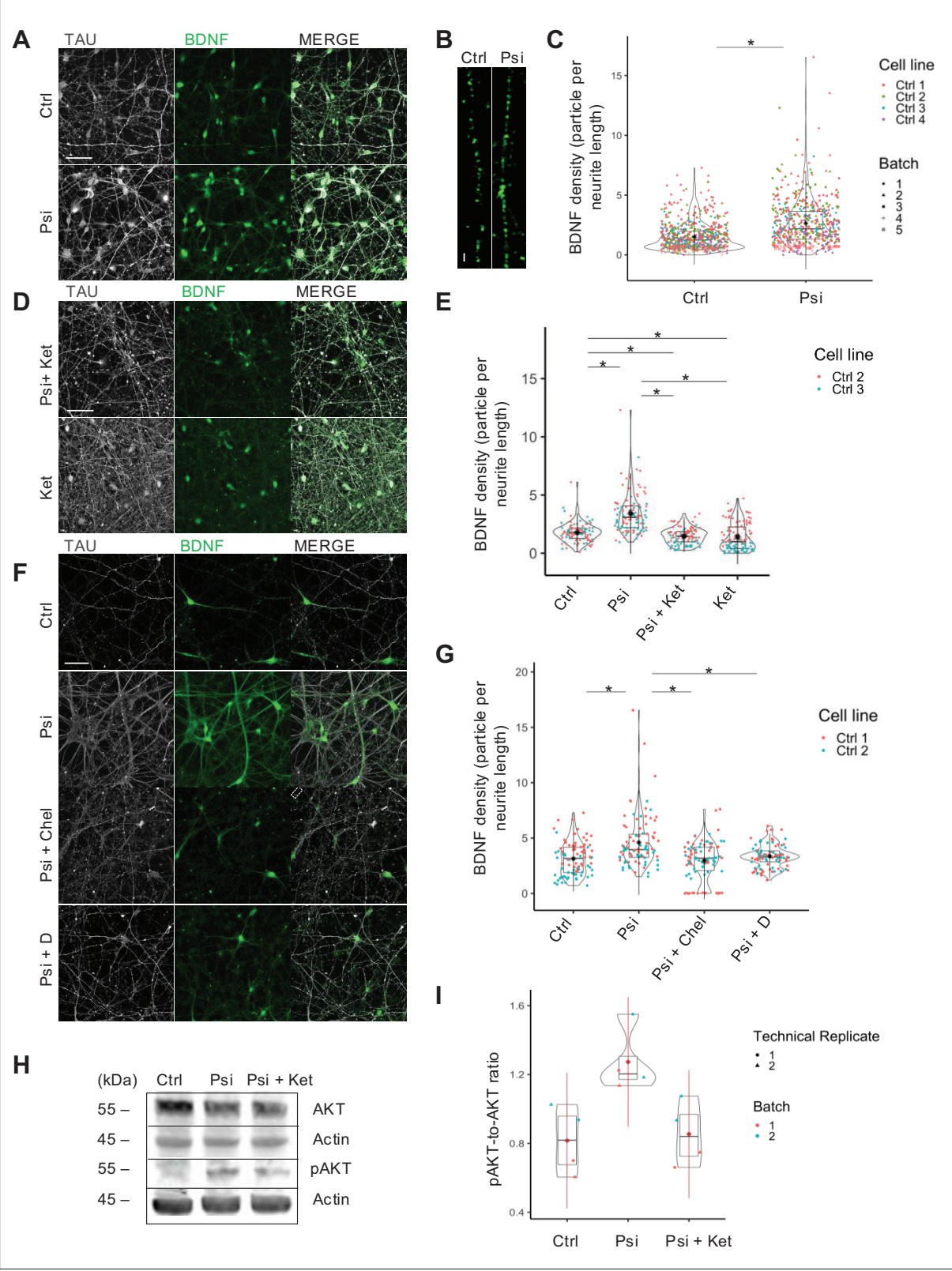

**Figure 2.** Psilocin-induced increase in BDNF level was 5-HT2A-R and PKC- and endocytosis-mediated and induced activation of m-BDNF/TrkB-associated downstream pathway. (**A, B**) Representative image of a neuronal network for pre-treatment condition (Ctrl) and 24 hrs after a 10 min 10 µM short psilocin trigger (Psi). Scale bar: 50 µm, (**B**) close-up: 2 µm. (**C**) BDNF density was significantly increased 24 hrs after a 10 min 10 µM psilocin trigger (Psi). Four Ctrl cell lines were included in the analysis (Ctrl with N=613 neurites; Psi with N=529 neurites). (**D**) Representative image 24 hrs after

*Figure 2 continued on next page*

*Figure 2 continued*

the simultaneous treatment with psilocin and ketanserin (Psi + Ket) and single treatment with ketanserin (Ket), scale bar: 50 µm. (**E**) Ketanserin co-treatment (Psi + Ket) and ketanserin monotreatment (Ket) significantly reduced BDNF density compared to the 24 hrs monotreatment psilocin condition, suggesting a 5-HT2A-R-mediated process. Ketanserin monotreatment provoked a significant reduction in BDNF density compared to ketanserin co-treatment with psilocin. Two Ctrl cell lines were included in the analysis, each with one biological batch (Ctrl with N=99 neurites; Psi with N=102 neurites; Psi + Ket with N=102 neurites; Ket with N=102 neurites). (**F**) Representative image 24 hrs after the simultaneous treatment of 10 µM psilocin with chelerythrine (Psi + Chel) or dynasore (Psi + D), scale bar: 50 µm. (**G**) Chelerythrine (Psi + Chel, selective PKC inhibitor) or dynasore (Psi + D, inhibition of clathrin-coated vesicle invagination) co-treatment significantly reduced BDNF density compared to the psilocin monotreatment condition (Psi). Two Ctrl cell lines were included in the analysis, each with one biological batch (Ctrl with N=90 neurites; Psi with N=90 neurites; Psi + Chel with N=90; Psi + D with N=90 neurites). (**H**) Phosphorylated AKT (pAKT) protein level was increased 72 hrs after psilocin exposure (Psi), reversed by ketanserin co-treatment (Psi + Ket), (**I**) both effects were not significant. Ctrl cell line 3 was included in the analysis (Ctrl with N=4 data points, Psi with N=4 data points, Psi + Ket with N=4 data points), each with two biological batches. For all analyses, the Kruskal–Wallis test for independent samples was calculated. Post hoc Wilcoxon rank sum test. Bonferroni correction, adjusted $p < .05$, mean ± SD. Significance levels against the respective control and for multiple group comparisons are *$p < 0.05$.

The online version of this article includes the following source data and figure supplement(s) for figure 2:

**Source data 1.** Western Blot: phosphorylated AKT (pAKT) protein level and AKT protein level after psilocin exposure (Psi) and psilocin and ketanserin treatment (Psi + Ket) (72 hrs post, uncropped, labeled).

**Source data 2.** Western Blot: phosphorylated AKT (pAKT) protein level and AKT protein level after psilocin exposure (Psi) and psilocin and ketanserin treatment (Psi + Ket) (72 hrs post, uncropped).

**Figure supplement 1.** Influence of different treatment conditions on BDNF level.

**Figure supplement 1—source data 1.** Western Blot: phosphorylated TrKB (pTrkB), pro-BDNF and m-BDNF protein level after psilocin exposure (Psi) (24 hrs post, uncropped, labeled).

**Figure supplement 1—source data 2.** Western Blot: phosphorylated TrKB (pTrkB), pro-BDNF and m-BDNF protein level after psilocin exposure (Psi) (24 hrs post, uncropped).

**Figure supplement 1—source data 3.** Western Blot: phosphorylated AKT (pAKT) protein level and AKT protein level after psilocin exposure (Psi) (24 hrs post, uncropped, labeled).

**Figure supplement 1—source data 4.** Western Blot: phosphorylated AKT (pAKT) protein level and AKT protein level after psilocin exposure (Psi) (24 hrs post, uncropped, labeled).

arbor of single mature neurons in these cultures, the cultures were transduced with adeno-associated vectors (AAVs) coding for mCherry under control of the CaMKIIa promoter (*Figure 4A–D*).

We observed an increase in primary neurites at 25 µm (significant at 48 hrs and a trend at 24 hrs from MCtrl = 4.1 ± 2.1; M24hrs = 5.4 ± 3.7; M48hrs = 5.5 ± 2.4) and of intersections at 50 µm (significantly increased at 24 and 48 hrs post treatment from MCtrl = 2.7 ± 1.4; M24hrs = 4 ± 2.3 to M48hrs = 3.9 ± 1.8; *Figure 4B*). As a result, the calculated total neurite length was significantly increased at 24 and 48 hrs post-treatment from MCtrl = 272.5 ± 142.7; M24hrs = 383.2 ± 209.9; M48hrs = 352.8 ± 162.5 (*Figure 4C*). And we observed a significant increase in the total number of intersections in psilocin-exposed neurons quantified in steps of 25 µm up to a distance of 125 µm (MCtrl = 10.9 ± 5.7; M24hrs = 15.3 ± 8.4, M48hrs = 14.1 ± 6.5; *Figure 4D*). These data indicate that gene expression changes elicited by psilocin resulted in neurite outgrowth and an increase of global dendritic complexity as early as 24 hrs following a single 10 min exposure.

## Psilocin increases synaptic strength and synaptogenesis

To address changes in neuronal function, we exposed neuronal cultures to 10 min or 24 hrs to 10 µM psilocin and performed whole-cell patch-clamp experiments one week later. Following the 10 min exposure, we observed an increasing trend in the number of evoked action potentials (APs), having amplitudes of around 100 mV (*Figure 5A*). Extended exposure of the cultures to psilocin (24 hrs) increased this effect, reaching significance (total number of APs from MCtrl = 20.1 ± 12; MPsi 10 min; day 7=28.9 ± 16.9; MPsi 24 hrs; day 7=30 ± 14.5; AP amplitude from MCtrl = 101.1 ± 13.3; MPsi 10 min; day 7=101.1 ± 13.2; MPsi 24 hrs; day 7=101.3 ± 16.7). To find out whether the increase in AP firing enhanced synaptic network activity, we recorded spontaneous AP-dependent excitatory postsynaptic currents (sEPSCs). Indeed, the frequency of sEPSCs increased in both psilocin-treated conditions to some extent (MCtrl = 0.6 ± 0.7; MPsi 10 min; day 7=1.0 ± 1.3; MPsi 24 hrs; day 7=1.1 ± 1.2; *Figure 5B*). And we observed a slight increase in sEPSC amplitude (MCtrl = 14.2 ± 4.7; MPsi 10 min; day 7=16.7 ± 5.1; MPsi 24 hrs; day 7=15.5 ± 5.3; *Figure 5B*), consistent with the observed

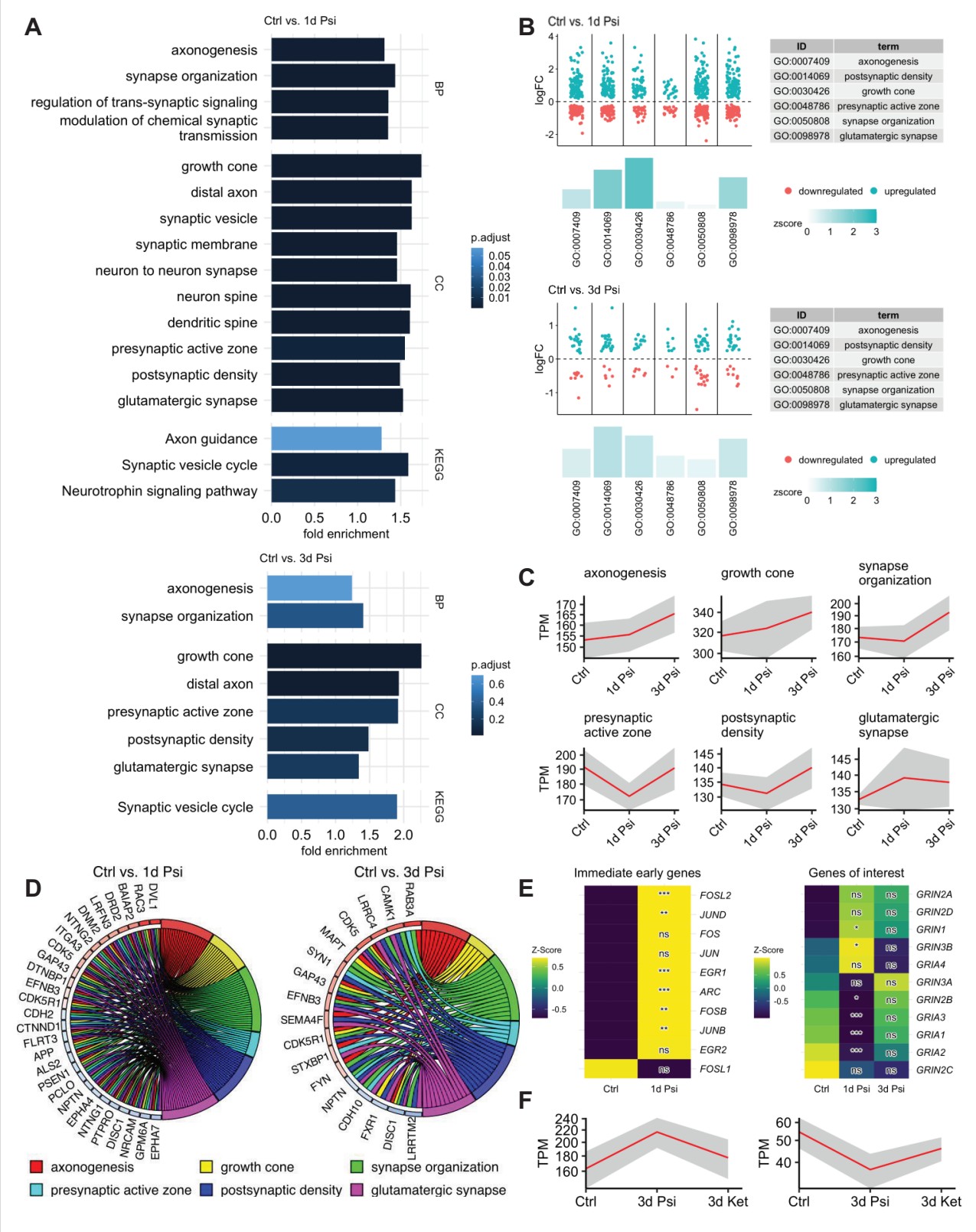

**Figure 3.** Psilocin displays fast and enduring changes of the genetic landscape. (**A**) Enrichment in differentially expressed genes associated with synapse formation, neuronal plasticity, and axonogenesis GO terms 1 day after psilocin administration, and the effect 3 days later. (**B**) Psilocin induced within 24 hrs a first wave of upregulation of selected GO genes based on DESeq2 normalized counts. Log2fold changes of each differentially expressed gene between two conditions are shown as dots. Z scores indicate if more genes in the respective GO term are upregulated or downregulated, also

*Figure 3 continued on next page*

*Figure 3 continued*

indicated by height and color of the bars. (**C**) TPM-normalized mean (red line) of psilocin-induced temporal expression pattern of genes belonging to the indicated GO. The shaded area indicates SD. (**D**) Chord plots showing significant genes appearing in at least four GO terms (of six selected GO terms) 1 day after psilocin administration and appearing in at least three GO terms 3 days after psilocin administration. (**E**). Heat map (z-scaled normalized counts) showing expression of immediate early genes and AMPA/NMDA receptor genes throughout psilocin administration. (**F**) Differentially expressed genes that are up-/downregulated upon psilocin treatment showed a reversed effect upon co-treatment with ketanserin. Mean TPM-normalized expression values are shown (red line), shaded area indicates S.D. BP, biological process; CC, cellular compartment; KEGG, Kyoto Encyclopedia of Genes and Genomes; TPM, transcripts per kilobase million. The significance was assessed using a Wald test. Significance levels against the respective control ns: p-adj.>0.05, *p-adj. ≤ 0.05, **p-adj. ≤ 0.01, ***p-adj. ≤ 0.001.

The online version of this article includes the following figure supplement(s) for figure 3:

**Figure supplement 1.** Psilocin displays fast and enduring changes on GO terms related to learning, memory, and cognition.

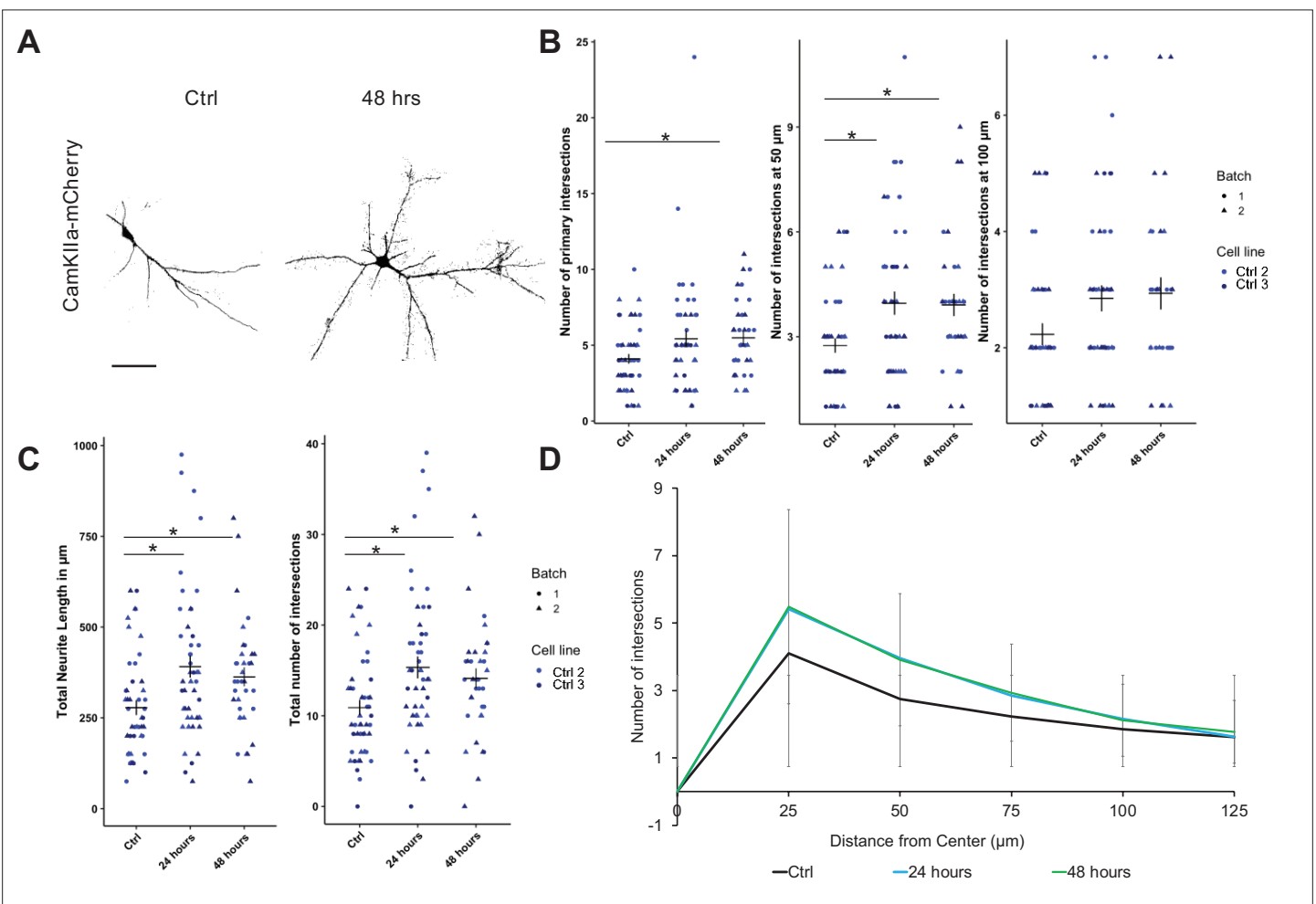

**Figure 4.** Psilocin-induced neurite branching. (**A–C**) Cells were transduced with AAV CamKIIa p-hCHR2(134a)-mCherry. (**A**) Representative image for pre-treatment condition (Ctrl) and 48 hrs after a 10 min short psilocin trigger (48 hrs) for mCherry staining showed an increase in neurite intersections for the latter. (**B**) Singular analysis: 48 hrs after a 10 min 10 µM psilocin trigger, the number of primary (25 µm distance from soma) neurite intersections significantly increased compared to the untreated control condition (Ctrl). The number of intersections at 50 µm distance from soma significantly increased 24 hrs and 48 hrs after a 10 min 10 µM psilocin trigger. (**C**) Significant changes for the calculated total neurite length and for the total number of neurite intersections significantly increased 24 hrs and 48 hrs after a 10 min 10 µM psilocin trigger. Two control cell lines with two biological batches (Ctrl with N=43–48 neurites; 24 hrs with N=47–48 neurites; 48 hrs with N=31–35 neurites) were included. (**D**) Sholl analyses summary for results shown in figure (**B**). For all analyses, the Kruskal–Wallis test for independent samples was calculated. Post hoc Wilcoxon rank sum test. Bonferroni correction, adjusted p<0.05, mean ± SD. Significance levels against the respective control are *p<0.05.

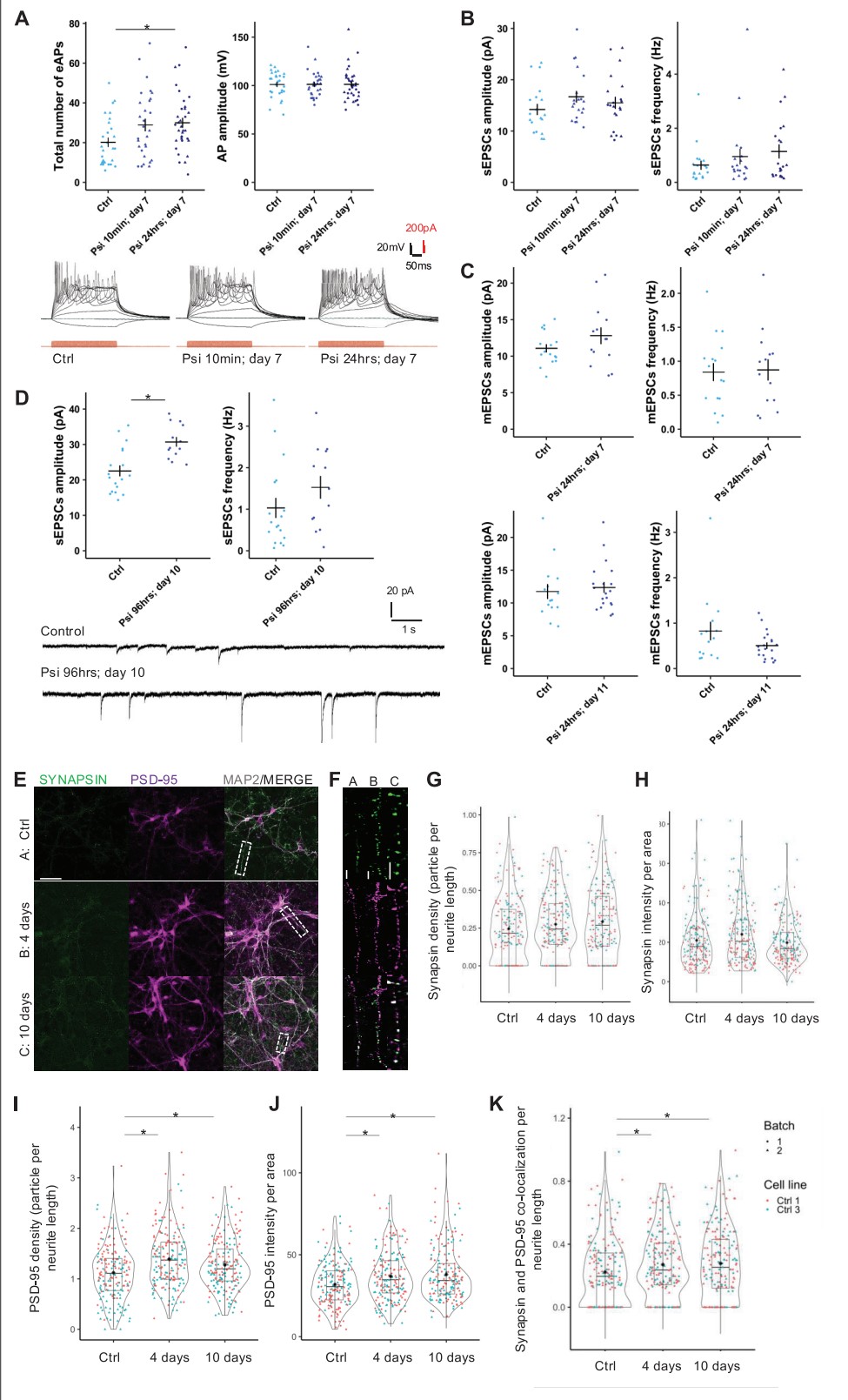

**Figure 5.** Psilocin-induced increase in synaptic strength and synaptogenesis. (**A**) Total number of evoked action potentials (eAPs) significantly increased at day 7 after 24 hrs permanent psilocin administration (Psi 24 hrs; day 7) and was increased 7 days after a 10 min psilocin trigger (Psi 10 min; day 7). AP amplitudes stayed constant. Representative traces for the total number of APs. One Ctrl cell line with 2 biological batches was included in the

*Figure 5 continued on next page*

*Figure 5 continued*

analysis. Ctrl 3 with N=28 cells, Psi 10 min; day 7 with N=29 cells, Psi 24 hrs; day 7 with N=34 cells. (**B**) Increase in sEPSCs amplitude and frequency 7 days after 10 min and 24 hrs permanent psilocin administration. One Ctrl cell line with 2 biological batches was included in the analysis, Ctrl 3 with N=19 cells, Psi 10 min; day 7 with N=20 cells, Psi 24 hrs; day 7 with N=21 cells. (**C**) Increase in mEPSCs amplitude 6 days after permanent 10 µM psilocin administration, Ctrl 3 with N=16 cells, Psi 24 hrs; day 7 with N=14 cells. Increase in mEPSCs amplitude 10 days after 24 hrs permanent psilocin administration, Ctrl 3 with N=15 cells, Psi 24 hrs; day 11 with N=20 cells. (**D**) Significant increase in sEPSCs amplitude 6 days after 96 hrs permanent 10 µM psilocin administration (Psi 96 hrs; day 10). Ctrl 3 with N=18 cells, Psi 96 hrs; day 10 with N=13 cells. Representative traces for control and psilocin conditions. For all experiments, the Mann–Whitney U-test for independent samples was calculated, mean ± SEM. Significance levels against the respective control are *p<0.05. (**E, F**) Representative dendritic PSD-95 and synapsin staining for the untreated condition (Ctrl), 4 days after permanent 10 µM psilocin administration (4 days) and 10 days after 4 days permanent 10 µM psilocin administration (10 days). (**E**) Scale bar: 50 µm (**F**), close-up: 10 µm. Trend in synapsin (**G**) density and (**H**) intensity increased 4 days after permanent psilocin administration. (**I**) PSD-95 particle per neurite length and (**J**) intensity per area and (**K**) synapsin/PSD-95 co-localization were significantly increased 4 and 10 days after 4 days permanent 10 µM psilocin treatment compared to an untreated control condition. (**G–K**) Ctrl with N=180 neurites, 4 days with N=180 neurites, 10 days with N=180 neurites. Two control cell lines, each with two biological batches, were included. For all analyses, the Kruskal–Wallis test for independent samples was calculated. Bonferroni correction, adjusted p<0.05, mean ± SD. Significance levels against the respective control are *p<0.05.

The online version of this article includes the following figure supplement(s) for figure 5:

**Figure supplement 1.** Psilocin-induced increase in synaptic strength.

---

*GRIA* upregulation (***Figure 3E***). The latter amplitude increase was next examined on AP-independent miniature EPSCs (mEPSCs), which are generally caused by the spontaneous release of single vesicles (***Figure 5C***, ***Figure 5—figure supplement 1A***). The amplitudes of mEPSCs increased 7 days after the start of 24 hrs psilocin administration consistently for two cell lines (MCtrl = 11.1 ± 2.2; MPsi 24 hrs; day 7=12.8 ± 4.2, ***Figure 5C***; MCtrl = 9.3 ± 2.8; MPsi 24 hrs; day 7=11.2 ± 3.7, ***Figure 5—figure supplement 1A***), whereas the frequencies of the mEPSCs were rather stable (MCtrl = 0.8 ± 0.5; MPsi 24 hrs; day 7=0.9 ± 0.6, ***Figure 5C***; MCtrl = 0.9 ± 0.7; MPsi 24 hrs; day 7=0.7 ± 0.5, Fig. ***Figure 5—figure supplement 1A***). When examining mEPSCs from separately differentiated neurons exposed to psilocin for 24 hrs and analyzed at day 11, the increase in mEPSC amplitude was somewhat attenuated compared with 7 days (MCtrl = 11.7 ± 4.3; MPsi 24 hrs; day 11=12.4 ± 3.6; frequency from MCtrl = 0.8 ± 0.8; MPsi 24 hrs; day 11=0.5 ± 0.3, ***Figure 5C***). Repeated LSD administration altered gene and protein expression related to neuroplasticity signaling in the mouse prefrontal cortex and increased dendritic spine density (***Inserra et al., 2022***; ***De Gregorio et al., 2022***). We therefore hypothesized that extended psilocin administration fosters synaptic network activity and synaptogenesis and thus examined effects of a 96 hrs exposure. In line with the above-mentioned hypothesis, the amplitudes of the sEPSCs increased significantly in two lines (MCtrl = 22.5 ± 6.5; MPsi 96 hrs; day 10=30.7 ± 4.8, ***Figure 5D***; MCtrl = 15.9 ± 4.5; MPsi 96 hrs; day 10=27.6 ± 10, ***Figure 5—figure supplement 1B***), and the frequencies showed a trend to increase similar to ***Figure 5B***, reflecting enhanced network activity (MCtrl = 1.0 ± 1.0; MPsi 96 hrs; day 10=1.5 ± 1.0, ***Figure 5D***; MCtrl = 1.5 ± 1.7; MPsi 96 hrs; day 10=1.0 ± 0.7, ***Figure 5—figure supplement 1B***).

This enhanced synaptic network activity should affect the abundance of either the presynaptic marker synapsin and/or the postsynaptic marker PSD-95 and may even affect their co-localization at day 4 and at day 10 following 4 day permanent 10 µM psilocin administration. In line with the above electrophysiological results for the 10-day psilocin exposure (***Figure 5D***) and the gene expression data, which indicated changes associated with synaptogenesis and synaptic plasticity (***Figure 3***), we observed for the pre-synaptic marker synapsin, a trend towards an increase of the density (particles per neurite length; from MCtrl = 0.25 ± 0.2; M4days = 0.28 ± 0.2; M10days = 0.29 ± 0.2) and intensity (mean fluorescence intensity of the particles; from MCtrl = 21 ± 13.3; M4days = 24.1 ± 15; M10days = 19.8 ± 10.9) at both time points (***Figure 5E–H***). In line with the gene expression data (***Figure 3A***, postsynaptic density), we observed a significant increase in the density of the postsynaptic marker PSD-95 (from MCtrl = 1.1 ± 0.5; M4days = 1.4 ± 0.6; M10days = 1.3 ± 0.5) and in its intensity (from MCtrl = 31.8 ± 12.9; M4days = 36.8 ± 15.6; M10days = 38 ± 16.4) for both time points (***Figure 5I and J***). These effects were accompanied by a significant increase in the co-localization of both markers

(from MCtrl = 0.2 ± 0.2; M4days = 0.3 ± 0.2; M10days = 0.3 ± 0.2; *Figure 5K*). Together, these experiments indicate that psilocin augments synaptic strength primarily via an increase in the postsynaptic receptor density, which lasts at least for 6 days from the start of psilocin withdrawal and that extended exposure of neurons to psilocin pronounces the effects on synaptic strength.

## Discussion

The neuroplasticity-promoting psychoplastogen (*Ly et al., 2018*) psilocybin is currently being developed as a new medication in the treatment of psychiatric disorders (*Nichols et al., 2017*; *Carhart-Harris and Goodwin, 2017*; *Lowe et al., 2021*). 'Brain network resetting' (*Nichols et al., 2017*) by psilocybin, that is, the restoration of neuronal and synaptic dysfunction associated with the pathophysiology of mental disorders (*Bhattacharyya et al., 2002*; *Kowiański et al., 2018*; *Moliner et al., 2023*; *de la Fuente Revenga et al., 2021*; *Wojtas et al., 2022*), could be achieved by stimulating signaling pathways associated with neuroplasticity and BDNF signaling (*de Vos et al., 2021*; *Kwan et al., 2022*). So far, experimental evidence for this hypothesis is based on data from non-human model organisms.

In our experimental setting, we observed a pronounced psilocin-induced BDNF upregulation in human cortical neurons and showed that PKC activation and endocytosis, two mechanisms contributing to 5-HT2A-R internalization (*Berry et al., 1996*; *Bhattacharyya et al., 2002*), are involved as

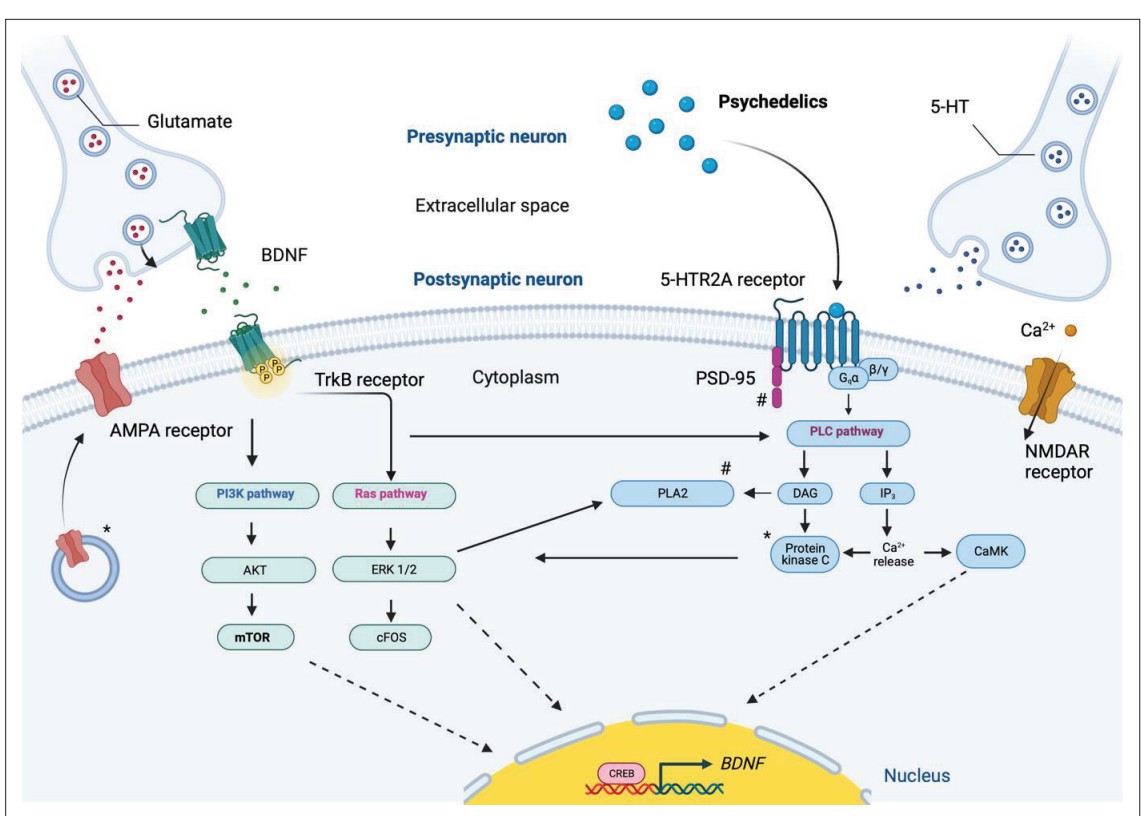

**Figure 6.** Simplified schematic of signal transduction pathways proposed to mediate psychedelic-induced neuroplasticity in the cortex. Serotonergic psychedelics activate the 5-HT2A-R, triggering G-protein–coupled PLC and PLA2 pathways, which, similar to m-BDNF-TrkB activation, increase intracellular $Ca^{2+}$ and PKC activity. BDNF–TrkB signaling further activates mTOR and NMDAR-dependent plasticity via the PI3K/AKT pathway. Psychedelic-induced glutamate release promotes mTOR activation and BDNF release through AMPARs, enhancing glutamatergic signaling and IEG (e.g., c-FOS) expression. 5-HT2A-R also interacts with PSD-95, while PI3K, ERK1/2, and CaMK activate CREB, stimulating *BDNF* transcription. Image created in BioRender, adapted from *de Vos et al., 2021* and used in *Schmidt et al., 2024*. 5-HT, serotonin; AKT, protein kinase B; AMPAR, α-amino-3-hydroxy-5-methyl-4-isoxazolepropionic acid receptor; BDNF, brain-derived neurotrophic factor; CaMK, $Ca^{2+}$/calmodulin-dependent kinase; CREB, cAMP response element–binding protein; ERK1/2, extracellular signal–regulated kinase 1/2; IEG, immediate early gene; MAPK, mitogen-activated protein kinase; mTOR, mammalian target of rapamycin; NMDAR, N-methyl-D-aspartate receptor; PI3K, phosphoinositide 3-kinase; PKC, protein kinase C; PLA2, phospholipase A2; PLC, phospholipase C; PSD-95, postsynaptic density protein 95; TrkB, tropomyosin receptor kinase B.

chelerythrine and dynasore inhibited BDNF augmentation. BDNF plays an important role in neurogenesis, synaptogenesis, and the formation of synaptic interactions, the basis of neuroplasticity (*Figure 6*), the brain's enduring ability to constantly change in response to influences and experiences and lifelong learning (*Kowiański et al., 2018*). Particularly, the mature BDNF/phosphorylated-TrkB receptor complex is involved in multiple pathways linked to prosurvival and synaptic plasticity (*Kowiański et al., 2018*). Recently, it was shown that psychedelics also directly bind to the BDNF TrKB receptor, thereby affecting TrkB dimerization and facilitating the effect of endogenous BDNF, which underlines the importance of the BDNF system for the action of psychedelics (*Moliner et al., 2023*). In keeping with this, we show that psilocin leads to a fast and enduring upregulation of proteins and genes linked to neuronal complexity, synaptogenesis and synaptic transmission, as also demonstrated in psychedelic-mediated plasticity in animal studies (*Ly et al., 2018*; *Shao et al., 2021*; *de la Fuente Revenga et al., 2021*). Of note, BDNF upregulation and changes in gene expression were reversed by ketanserin, confirming that these effects are mediated by 5-HT2A-Rs.

In our model, augmentation of neuronal complexity is an outcome which can be detected as early as 24 hrs after psilocin administration. In rodents, serotonergic psychedelics also foster synaptogenesis and spinogenesis (*Ly et al., 2018*; *Shao et al., 2021*; *de la Fuente Revenga et al., 2021*). In line with this, we show that psilocin also promotes synaptogenesis, measured by the increase in PSD-95, synapsin and their co-localization. As a result, we observe changes in intrinsic neuronal properties and network function. More specifically, we see an increased excitability and an increase in postsynaptic current frequency but, in particular, amplitude. The higher number of action potentials generated by current injections could be due to increased dendritic excitability, as reviewed by *Kwan et al., 2022*. The increase in synaptic strength lasted at least 6 days and is in agreement with frequency and amplitude increases of miniature and spontaneous synaptic currents observed in acute brain slices of mice after administering psilocybin or DMT *in vivo* (*Ly et al., 2018*; *Shao et al., 2021*). These drugs may increase extracellular glutamate levels, similar to ketamine, LSD and DOI administration (*de Vos et al., 2021*; *Wojtas et al., 2022*; *Mason et al., 2020*; *Muschamp et al., 2004*; *Scruggs et al., 2003*). To close the circle, 5-HT2A-R stimulation, subsequent glutamate release and AMPA receptor activation (*Lowe et al., 2021*) activate the BDNF-associated TrkB and mTORC1 pathway which promotes BDNF expression itself (*Duman et al., 2016*; *Vollenweider and Kometer, 2010*). Of note, PSD-95, the postsynaptic marker that we found to be upregulated, controls activity-dependent AMPA receptor incorporation in the postsynapse (*Ehrlich and Malinow, 2004*; *Opazo et al., 2012*), which can modify the strength of excitatory synaptic transmission (*Chater and Goda, 2014*).

Finally, an enrichment of differentially expressed genes associated with GO terms of learning, memory, and cognition suggests behavioral long-term effects of the drug which are based on the above-mentioned structural and functional modifications. The effects were dependent on the duration of drug exposure, suggesting that repetitive administration of psilocybin might elicit beneficial effects with respect to brain plasticity.

Our study has some limitations. The *in vitro* system lacks metabolizing enzymes and the cellular complexity of brain tissue including glial cells. Furthermore, some experiments required prolonged psilocin exposure (96 hrs), which does not fully recapitulate the *in vivo* situation where psilocin is rapidly metabolized. Nevertheless, the majority of our key findings (BDNF upregulation, gene expression changes, neurite branching) were observed 24–48 hrs after a brief 10 min exposure, more closely reflecting the post-acute effects observed *in vivo*.

Together, our work confirms the postulation by the Olson group that psychedelics may act through an evolutionarily conserved mechanism, as we can replicate the results from animal studies in our human system (*Ly et al., 2018*). Our study underscores the significance of human-based models, particularly iPSC-derived cortical neurons, in elucidating the mechanisms of psychedelics and their potential for disease modeling and drug development. By leveraging these human-derived systems, the research emphasizes the translational potential of psychedelics, offering deeper insights into their action in a human context and advancing therapeutic innovation (*Liao et al., 2025*).

# Materials and methods

## Human material and generation of fibroblast-derived human iPSCs

The study was approved by the local ethics committee (Ethics Committee II of Heidelberg University; approval no. 2009-350N-MA). All experiments with human material were in accordance with the Declaration of Helsinki. All healthy female participants gave written informed consent. One hiPSC line is registered at https://hpscreg.eu/ (CIMHi001-A). Reprogramming was done using CytoTune-iPS 2.1 Sendai Reprogramming Kit (Thermo Fisher Scientific) from skin fibroblasts. Cells were cultured at 37°C, ambient $O_2$ and 5% $CO_2$ concentration under sterile conditions in an incubator (Binder). Applications with the cells were carried out in a sterile flow hood (Scanlaf Mars) by using sterile and autoclaved instruments and medium. Chromosomal alterations were excluded by genome-wide single nucleotide polymorphism (SNP) analysis. Mycoplasma testing was performed on a regular basis, and genetic identity was verified by DNA fingerprinting. Therefore, genomic DNA was extracted from hiPSCs and analyzed for polymorphic ALU insertions with the following 12 primer pairs: Al-1a, Al-2a, Al-3a, Al-4a, Al-5a, Al-6a, Al-7a, Al-8a, Al-9a, Al-10, Al-11, Al-12. ALU insertions primers based on *Mamedov et al., 2010* are listed in *Supplementary file 1a*. iPSCs were screened for their stem cell properties by the capability of *in vitro* differentiation into the three embryonic germ layers, ectoderm, mesoderm and endoderm, and for pluripotent stem cell markers *in vitro*.

## iPS cell culture

Fibroblast-derived iPS cells were kept as colonies under feeder-free conditions in stem cell state on 5% (v/v) Geltrex-coated plates (Thermo Fisher Scientific) containing 1% (v/v) Pen/Strep (Thermo Fisher Scientific) in Essential 8 medium (DMEM/F12 with L-glutamine and HEPES [Thermo Fisher Scientific] supplemented with 1% [v/v] Pen/Strep [v/v], 64 μg/ml LAAP [Sigma-Aldrich], 14 ng/ml sodium selenite [Sigma-Aldrich], 200 ng/ml FGF-2 [154; Cell Guidance Systems], 2 ng/ml TGF-β1 [Cell Guidance Systems], 20 μg/ml insulin [Sigma-Aldrich], 11 μg/ml transferrin [Sigma-Aldrich]) that was changed daily. For passaging, confluent iPS colonies were washed twice with phosphate-buffered saline (DPBS) and then incubated with 0.5 mM EDTA (Thermo Fisher Scientific) for 5–10 min at RT until the colonies started to detach. After aspirating the EDTA solution, the fractured colonies were gently resuspended in E8 medium containing 5 μM Rho-associated protein kinase (ROCK) inhibitor (Cell Guidance Systems).

## Differentiation of iPS cells into neuronal progenitors

For *in vitro* differentiation into neural progenitors, E8 medium was replaced by neural progenitor induction medium phase 1 (advanced DMEM/F-12 medium with glutamine [Thermo Fisher Scientific], with 1% [v/v] Pen/Strep [v/v], 2 mM GlutaMAX [Thermo Fisher Scientific], 1% [v/v] B-27 supplement with RA [Thermo Fisher Scientific], 10 μM SB-431542 [Cell Guidance Systems], 1 μM LDN-193189 [Stemcell Technologies], 2 μM XAV939 [Cell Guidance Systems], and 5 μM cyclopamine [Cayman Chemical]) when iPS cells reached 70–80% confluency. At day 4, cells were dissociated 1:2 to a monolayer, continue using the aforementioned medium. Upon day 8, cells were dissociated 1:2 and medium was changed to induction medium phase 2 (advanced DMEM/F-12 medium with glutamine with 1% [v/v] Pen/Strep, 2 mM GlutaMAX, 1% [v/v] B-27 supplement with RA, 200 nM LDN-193189) for 8 more days. Finally, the medium was changed to induction medium phase 3 (advanced DMEM/F-12 medium with glutamine with 1% [v/v] Pen/Strep, 2 mM GlutaMAX, 1% [v/v] B-27 supplement with RA, 20 ng/ml FGF-2 [147; Cell Guidance Systems]). Dissociation into single cells was done using TrypLE (Thermo Fisher Scientific) for 5–15 min at 37°C until the cells started to detach. Cells were then transferred with wash medium to a 15 ml tube and centrifuged for 4 min at 1000 × *g*. The pellet was gently resuspended in corresponding neural progenitor induction medium supplemented with 5 μM ROCK inhibitor.

## *In vitro* differentiation of neural progenitors into human cortical neurons

*In vitro* differentiation into human cortical neurons was launched by the change to neural differentiation medium phase 1, defined as day 0 of differentiation. Neural differentiation phase 1 medium consisted of neural base medium 1 (DMEM/F-12 medium with glutamine, 0.1% [v/v] GlutaMAX,

1.8 mM CaCl$_2$ [Sigma-Aldrich], 1% [v/v] Pen/Strep, 1% [v/v] B-27, 0.5% [v/v] N2 supplement [prepared in the laboratory as follows: DMEM/F-12 with glutamine, 500 µg/ml insulin, 10 mg/ml transferrin, 520 ng/ml sodium selenite, 1.611 mg/ml putrescine [Sigma-Aldrich] and 630 ng/ml progesterone [Sigma-Aldrich]], 1% [v/v] NEAA [Thermo Fisher Scientific], 1.6 mg/ml glucose [Carl Roth] supplemented with 200 µM ascorbic acid [Sigma-Aldrich], 1 µM LM22A [Sigma-Aldrich], 1 µM LM22B [Tocris Bioscience], 2 µM PD-0332991 [Selleckchem], 5 µM DAPT [Cell Guidance Systems]) and was applied until day 3 of differentiation. The cells were split using TrypLE into phase 2 medium on polyethylenimine (Sigma-Aldrich)/laminin (Sigma-Aldrich)-coated plates (1:2000 of 1% [PEI] in 25 mM boric acid [pH 8.4; Thermo Fisher Scientific]; 3.75 µg/ml laminin [Sigma-Aldrich] in DPBS [Thermo Fisher Scientific]). Neural differentiation medium phase 2–4 media consisted of neural base medium 2 with Neurobasal (Thermo Fisher Scientific) (plus 1.6 mg/ml glucose, 1% [v/v] Pen/Strep, 0.1% [v/v] GlutaMAX and 1% [v/v] B-27) and was supplemented additionally to supplements of phase 1 medium with 3 µM CHIR99021 (Cell Guidance Systems), 10 µM forskolin (Cell Guidance Systems) and 300 µM GABA (Sigma-Aldrich). At day 10 of differentiation, the medium was changed to medium phase 3 containing, additionally to supplements of phase 1 medium, 3 µM CHIR99021. From day 17 on, CHIR99021 was removed in phase 4 and the medium contained 200 µM ascorbic acid, 1 µM LM22A, 1 µM LM22B and 2 µM PD-0332991. At day 24, the medium was changed to neural differentiation phase 5 medium in neural base medium 3 (advanced MEM (Thermo Fisher Scientific), 1.6 mg/ml glucose, 1% (v/v) Pen/Strep, 0.1% (v/v) GlutaMAX, 1% (v/v) B-27) and 0.27 nM Bryostatin 1 (Merck Millipore) for the next 14 days. Medium was changed every 3–4 days half.

## Co-culture of human cortical neurons with mouse astrocytes

For electrophysiological experiments, mouse astrocytes were plated in a density of around 80.000 cells/well on PEI/laminin-coated coverslips on 24-well plates. Mouse astrocytes were cultured in astrocyte standard medium, consisting of astrocyte base medium (1% [v/v] Pen/Strep, 0.5% [v/v] N2, 1.6 mg/ml glucose) supplemented with 0.5% (v/v) N2, 1.6 mg/ml glucose, 0.1% B-27, 100 ng/µl epidermal growth factor (Cell Guidance Systems), 10 ng/ml FGF-2 (147) until they reached 80–90% density. Medium was changed every other day. 3 days before phase 1, neurons were split in a density of 200.000 cells onto the astrocytes, and the medium was changed to astrocyte differentiation medium, consisting of astrocyte base medium and 0.5% (v/v) N2, 1.6 mg/ml glucose and 10 ng/µl BMP-4 (Miltenyi Biotec). Medium was changed daily. Since the moment the cortical neurons were plated on the astrocytes, the medium was changed to neural differentiation medium and differentiation was continued as described.

## Neuroplasticity experiments in cortical neurons

For immunostainings, around 200,000 cells/well (50,000 for Sholl analysis) were plated onto coverslips in 24-well plates. For WB, DNA- and RNA-based experiments around 2–6 million cells/well were plated on 6-well plate dishes or 10 cm dishes. Treatment of cortical neurons started at day 42 of differentiation. 3 days prior to experiments, for drug dissolving and during the course of the experiment, the culture medium was replaced with neural base medium 3 containing 1% (v/v) Pen/Strep, 1.6 mg/ml glucose, 0.1% (v/v) GlutMAX, 0.1% (v/v) B-27, and 200 µM ascorbic acid to avoid distorting effects of small molecules on cellular signal transduction pathways. For 7 days long-term treatment cell medium changes twice a week were performed with neural differentiation phase 5 medium.

## Treatment of cortical neurons with psilocin

10 mM DMSO stock solution of light-sensitive psilocin was diluted 1:1000 (final DMSO concentration = 0.001%) to a final concentration of 10 µM psilocin (THC-Pharm). For the short trigger, mature cortical neurons were treated for 10 min with psilocin, washed once with advanced MEM containing 1% (v/v) Pen/Strep and were further cultured and then fixed or harvested after indicated time points post treatment. For the permanent treatment condition, cells were treated for 24 (96) hrs permanently with 10 µM psilocin. For the 96 hrs plus 6 days washout condition, cells were treated for 96 hrs permanently with 10 µM psilocin (fresh psilocin was replaced once), washed once and were further cultured in neural differentiation phase 5 medium.

## Treatment of cortical neurons with ketanserin

To determine whether psilocin-induced neuroplasticity effects were 5-HT2A-R-dependent, 75 mM DMSO stock solution of ketanserin (ApexBio) was diluted 1:1000 to a final concentration of 75 µM ketanserin (final DMSO concentration = 0.001%). Cells were pretreated with 75 µM ketanserin for 10 min to block the 5-HT2A-R, then treated for another 10 min with psilocin and ketanserin (final psilocin concentration = 10 µM, final ketanserin concentration = 75 µM, final DMSO concentration = 0.002%), washed once with advanced MEM containing 1% (v/v) Pen/Strep and further cultured for 24 hrs post treatment.

## Treatment of cortical neurons with chelerythrine

To investigate whether psilocin-induced neuroplasticity effects were PKC-mediated, a 1 mM DMSO stock solution of chelerythrine (Cayman Chemical) was diluted 1:1000 to a final concentration of 1 µM chelerythrine (final DMSO concentration = 0.001%). Cells were treated for 10 min with psilocin and chelerythrine (final psilocin concentration = 10 µM, final chelerythrine concentration = 1 µM, final DMSO concentration = 0.002%), washed once with advanced MEM containing 1% (v/v) Pen/Strep and were further cultured and then fixed after 24 hrs post treatment.

## Treatment of cortical neurons with dynasore

To determine whether psilocin-induced neuroplasticity effects were endocytosis-mediated, 100 mM DMSO stock solution of dynasore (Cayman Chemical) was diluted 1:2000 to a final concentration of 50 µM dynasore (final DMSO concentration = 0.0005%). Cells were pretreated with dynasore for 50 min to block endocytosis, then treated for another 10 min with psilocin (final psilocin concentration = 10 µM) and dynasore (final dynasore concentration = 50 µM; final DMSO concentration = 0.0015%), washed once with advanced MEM containing 1% (v/v) Pen/Strep and further cultured and then fixed after 24 hrs post treatment.

## Immunocytochemistry

After a first washing with PBS, cells were fixed in 4% PFA (Sigma-Aldrich), washed three times with PBS and blocked and/or permeabilized in blocking solution containing 10% FBS (Thermo Fisher Scientific) in PBS. Blocking was performed for 1 hr. Dilution of primary and secondary antibodies was performed in blocking solution. Please refer to *Supplementary file 1b*: Primary antibodies used for immunocytochemistry (ICC) for blocking/permeabilization solution and dilution of primary antibodies. Primary antibodies were incubated overnight at 4°C. Samples were washed three times with corresponding blocking solution. Secondary antibodies, conjugated to Alexa Fluor 488, 568, or 647 (Thermo Fisher Scientific), were diluted 1:1000 and were applied for 1 hr at room temperature (*Supplementary file 1c*: Secondary antibodies used for ICC). Samples were then washed once in PBS to remove unbound antibodies. Counterstaining of cell nuclei was carried out by using 300 nM DAPI (*Supplementary file 1d*: Fluorescent probe), incubated for 5 min at room temperature, and washed three times in PBS and once with ddH$_2$O. Slides were mounted with mounting solution (100 mM Tris-HCl [pH 8.5], 25% glycerol, 10% mowiol [Carl Roth], 0.6% DABCO [Carl Roth]) on glass coverslip and air-dried overnight. Cells were observed with the Inverted Leica DMIL LED Microscope. Stem cell and neural progenitor properties were imaged with the Leica DM6 B microscope with Thunder imaging software. For neuroplasticity experiments, a Leica confocal TCS SP5 II microscope was used. Brightfield images were made with a Fluorescence Microscope Celldiscoverer 7 microscope from Zeiss. For the analysis of immunofluorescence, a polygon selection of ROIs was chosen. The protein density and co-localization of particles was measured with the ComDet v.0.5.3 plug-in for ImageJ (National Institutes of Health [NIH], open source). A particle size (depending on experimenters' evaluation), constant for overall experiments (BDNF: 1.0; PSD-95: 3.0; Synapsin: 3.0) and an intensity threshold (in SD), adjusted within each experiment, was defined. For co-localization analysis, the channels of proteins that have to be co-localized were merged in advance. As the maximal distance between co-localized spots, 4.00 pixels were stated. The number of particles was divided by the total length of the neurite in µm measured with the straight-line tool spanning the polygonal ROI. Multiple ROIs were set per image. N=1 corresponds to one neurite segment detected by one ROI.

## Analysis of neuronal complexity

For Sholl analysis, around 50.000 cells/well were cultured on 24-well PEI/laminin-coated coverslips. Cells were transduced 2 weeks before psilocin treatment with adeno-associated virus AAV_CamKIIa

p-hCHR2(134a)-mCherry (kindly provided by AG Grinevich, Central Institute of Mental Health, Mannheim, Germany), when mCherry expression was visible by microscope. Cells were fixed after indicated treatment timepoints. Images were analyzed using the Sholl analysis plug-in of ImageJ (circle radii). Start radius: 25 µm, step size: 25 µm, end radius: 125 µm, set center from active ROI, preview. Crossings were calculated manually. Based on that, total length and sum of intersections were calculated. N=1 corresponds to one neuron.

## Protein isolation and measurement of protein concentration

Cells (1 well of 6-well plate) were washed with ice-cold DPBS, harvested by using a cell lifter and were transferred into a 1.5 ml tube on ice. In the following, the cells were centrifuged for 5 min at 5000 × $g$ at 4°C. The supernatant was removed and the sample was stored at -20°C. For protein isolation, cells were resuspended in 150 µl of protein lysis buffer (50 mM Tris-HCl [pH 7.4], 150 mM NaCl, 25 mM EDTA, 1% [v/v] SDS with one protease and one phosphatase inhibitor mini tablet [Thermo Fisher Scientific] per 10 ml of protein solution) and incubated for 10 min at room temperature and then incubated for 1 hr on ice. To shear gDNA and reduce viscosity, samples were sonicated (20% duty cycles, 50% output, five pulses) from a Branson Ultrasonics sonifier 250 (Thermo Fisher Scientific). Samples were centrifuged for 5 min for 5000 × $g$ at 4°C. Supernatant was transferred to a new tube. Protein concentration was measured with the BCA protein assay (Thermo Fisher Scientific) following the manufacturer's instructions. Samples were diluted 1:5 in ddH$_2$O, and the absorption at 562 nm was measured in a PowerWave XS microplate reader (BioTek). The protein concentration of samples was calculated based on the included BSA standard dilution series.

## SDS-polyacrylamide gel electrophoresis and western immunoblotting

15–30 µg of protein was mixed with 6x denaturing protein sample buffer (93.75 mM Tris-HCl [pH 6.8], 6% SDS, 6% glycerol, 9% 2-mercaptoethanol, 0.25% bromophenol blue) and boiled for 5 min at 95°C. For the separation of proteins,, an SDS-PAGE (Bio-Rad Laboratories) was performed in a Mini-PROTEAN 2-D electrophoresis cell chamber and with the PowerPac Basic Power Supply. First, the polyacrylamide gel (Gel buffer for polyacrylamide gels, SDS-PAGE: 3 M Tris-HCl [pH 8.5], 0.3% [w/v] SDS; SDS-Polyacrylamide, stacking gel: 24.8% [v/v] Gel buffer, SDS-PAGE, 3.84% [v/v] Bis/acrylamide, 0.00672% [w/v] APS, 0.224% TEMED; SDS-Polyacrylamide, separating gel: 33.3% [v/v] Gel buffer, SDS-PAGE, 10% [v/v] Bis/acrylamide, 10% [v/v] glycerol, 0.028% [w/v] APS, 0.09% [v/v] TEMED) runs for about 30 min with 30 V for stacking of proteins and then for about 1.5–2 hrs at 110 V. A semi-dry blotting was performed using the Trans-Blot Turbo transfer system (Bio-Rad Laboratories) performed with 20 V and 1 A for 30–60 min. A reference protein marker (PS10 plus, GeneON Bioscience) was included to determine the size of the protein bands. For blotting, 0.45 µm pore size nitrocellulose blotting membrane (Sigma-Aldrich) was used. The membrane was surrounded from both sides by six layers of WB filter tissue (VWR). Filter tissue and membrane were wetted in transfer buffer (10% [v/v] Tris-glycine buffer, 20% [v/v] methanol, 0.08% [v/v] SDS). Membranes were blocked for 60 min in 5% BSA (Sigma-Aldrich) in TBS-T (10% [v/v] of TBS with 248 mM Tris-HCl [pH 7.4], 1.37 M NaCl, 26.8 mM KCl plus 0.1% [v/v] Tween20 [Sigma-Aldrich]) in a 50 ml tube. Primary antibodies used for western blotting (*Supplementary file 1e*) were diluted in 5% BSA in TBS-T and were incubated overnight at 4°C or for 1 hr at room temperature on a rolling mixer. The membrane was washed three times in TBS-T for 10 min at room temperature on the next day. 1:15,000 infrared DyLight IR-dye conjugated secondary antibodies used for WB (*Supplementary file 1f*) in TBS-T were applied for 1 hr at room temperature. Washing in TBS-T was carried out three times and once in TBS for 10 min at room temperature. Visualisation of tagged proteins was carried out with the Odyssey IR WB imaging system (Li-COR). Signals were normalized by 1: 20.000 β-actin levels. For quantification, regions of interest (ROIs) were set around lanes and intensity was measured with the densitometry analysis in ImageJ.

## Ribonucleic acid isolation

For ribonucleic acid (RNA) isolation, cells were (2 wells of 6-well plate) resuspended in 500 µl peqGold Trifast (VWR) and incubated for 10 min at room temperature. 100 µl of chloroform (Sigma-Aldrich) was added to the lysate and incubated for 10 min at room temperature. Tubes were then centrifuged for 5 min at 12,000 × $g$ at 4°C. The upper clear, aqueous nucleic acid phase was transferred into a new tube. 250 µl of isopropanol (Th. Geyer) was added to each sample. For RNA precipitation, tubes were

kept overnight at -20°C. Then, tubes were centrifuged for 15 min at 4°C of 12,000 × $g$ and supernatant was discarded. The pellet was washed twice with 75% ethanol (Th. Geyer) in DEPC water (Carl Roth) and centrifuged for 10 min at 4°C at 12,000 × $g$. After the last washing step, all ethanol had to evaporate by first discarding the ethanol and afterwards letting air-dry the pellet for about 30 min. The pellet was resuspended in 20 µl DEPC-treated $H_2O$ and shaken at 400 rpm at 37°C. To prevent DNA contamination, the DNAase I Amplification grade kit (Sigma-Aldrich) was used.

## Synthesis of complementary DNA

Complementary DNA (cDNA) was synthesized via the iScript cDNA synthesis kit (Bio-Rad Laboratories) according to manufacturer's instructions. 500 ng to 1 µg cDNA was used to reversely transcribe RNA into cDNA. Standard cycling program for cDNA synthesis: 25°C for 5 min, 46°C for 20 min, 95°C for 1 min. The resulting cDNA was diluted 1:5 in nuclease-free $H_2O$. (1x) Taq reaction buffer (Biozym), each 200 µM 10 mM dNTP mix (Steinbrenner Laborsysteme), each 400 nM 10 µM primer mix, 0.625 U Taq DNA polymerase (5 U/µl; Biozym), 0.4–10 ng Template DNA. Standard RT-PCR cycling program: Samples were denatured by heating at 95°C for 1 min followed by 35 cycles of amplification and quantification (95°C for 15 s and 60°C for 15 s, and 72°C for 10 s) and by a final extension cycle (72°C, 5 min). Primers for real-time (RT)-polymerase chain reaction (PCR) are listed in *Supplementary file 1g* (Primers for RT-PCR). For expression control, total human adult (BioCat) and fetal brain (Tebu Bio) were included.

## Agarose gel electrophoresis

Samples were mixed with 10x DNA sample buffer (50 mM Tris-HCl, pH 7.6, 0.25% [w/v] bromphenol blue, 60% glycerol). DNA fragments were separated in a 1–2% (w/v) agarose gel in 1x TAE buffer (40 mM Tris, 20 mM acetic acid, 1 mM EDTA), containing 1: 15,000 peqGreen (VWR) for staining of DNA. A reference DNA marker (100 bp or 1 kbp, New England Biolabs) was included to determine the size of the amplicons. Gels ran for 50 min at 100 V and were imaged on a GeneFlash imaging system (Syngene).

## RNA bulk sequencing and analysis

For each sample, 30 µl of RNA solution (60 ng/µl) in $ddH_2O$ was sent to the High Throughput Sequencing (seq) Unit of the Genomics & Proteomics Core Facility at the DKFZ (Heidelberg, Germany) to be processed for RNA Bulk Sequencing. Samples were run through in-house quality control, and only samples with an RNA integrity number (RIN)≥5.0 were used for cDNA library preparation according to the TruSeq Stranded protocol (Illumina). Libraries were sequenced to 50 bp paired reads on the Illumina NovaSeq 6K platform to an average of 40M reads per sample. The DKFZ Omics IT and Data Management Core Facility performed the RNAseq processing workflow. Total counts per feature were imported to R (*R Development Core Team, 2020*) (v. 4.0.3–4.2.2) and analysed using DESeq2 (*Love et al., 2014*) (v. 1.30.1–1.38.3). All features without any counts were removed, for differential testing with DESeq2, the formula '~Batch + Cell_line +Condition_simple' was used. Differentially expressed genes (padj ≤ .05) were used to perform gene ontology (GO) enrichment analysis and Kyoto Encyclopedia of Genes and Genomes (KEGG) enrichment analysis with the clusterProfiler v 4.8.3 package (*Yu et al., 2012*). The organism database used for this analysis was org.Hs.eg.db (*Carlson, 2026*) (v. 3.17.0). GO Chord Plots were created using GOplot (v. 1.0.2) (*Walter et al., 2015*).

Z scores in GO plots showing up/downregulated genes were calculated with the following formula: zscore=(up-down)/sqrt(count). Heatmaps for RNAseq expression data show z-scaled DESeq2 normalized counts. Biological batch replicates of each sample ensured intra-cell line stability of results. The R code used to analyze RNAseq data is available at https://github.com/ahoffrichter/Schmidt_et_al_2025 (copy archived at *Hoffrichter, 2026*).

## Electrophysiology

Whole-cell patch-clamp recordings from iPSC-derived cortical neurons on PEI/laminin-coated coverslips cultured on mouse astrocyte cells were made using an EPC9 amplifier and PatchMaster (HEKA Elektronik GmbH). The neurons were identified with a Zeiss Axioskop using infrared differential interference contrast video microscopy in the recording chamber. Neurons were perfused at 2 ml/min (peristaltic pump; Ismatec GmbH) with carbogen (95% $O_2$; 5% $CO_2$)-saturated artificial cerebrospinal

fluid (ACSF) containing 125 mM NaCl, 1 mM $MgCl_2$, 2 mM $CaCl_2$, 2.5 mM KCl, 10 mM D-glucose, 25 mM $NaHCO_3$, 1.25 mM $NaH_2PO_4$ (pH 7.3, osmolarity 300 mOsm). Pipettes were pulled from borosilicate glass capillaries (GB150F-8P, 1.5 mm o.d., 0.86 i.d.; Science Products GmbH) with a P-97 micropipette puller (Sutter Instrument). Pipettes had resistances between 4 and 6 MΩ when filled with intracellular solution containing 115 mM K-gluconate (or cesium-gluconate), 20 mM KCl, 10 mM Na-phosphocreatine, 4 mM Mg-ATP, 0.3 mM GTP, 0.2 mM ethylene glycol tetraacetic acid (EGTA), and 10 mM HEPES (pH 7.3, osmolarity 300 mOsm). In voltage-clamp, input resistance was determined at –70 mV based on currents evoked by small voltage steps (–3 mV; 300 ms). In current-clamp, the resting membrane potential (RMP) was determined and APs were evoked with increasing current injections (10 pA; 300 ms). The amplitude of the first evoked AP was analyzed, and the number of all APs evoked by 10 depolarizing steps was summed up. sEPSCs were recorded at –70 mV with K (*Figure 5B*) or Cs (*Figure 5D*, *Figure 5—figure supplement 1B*)-based intracellular solution. Miniature EPSCs (mEPSCs) were recorded at –70 mV in 1 µM TTX and with Cs-based intracellular solution. Recordings were made at room temperature and were sampled at 20 kHz. Offline analysis was done with FitMaster (HEKA Elektronik GmbH). Spontaneous synaptic activity was analyzed using MiniAnalysis (Synaptosoft).

## Statistical analysis visualization

Unless indicated otherwise, data for quantitative analysis was based on at least two genetically independent cell lines with two independent biological replicates. For statistical analysis, the RStudio for macOS (version 1.4.1106 2009) was used. Graphs were generated with RStudio and show mean including single data points ± standard error of mean (SEM) or violin plots including single data points (scatterplot) with mean ± standard deviation (SD) and boxplot with median. The Kolmogorov–Smirnov test for equality of a probability distribution and the Levene test for homogeneity of variance were calculated prior to statistical analysis. In case the data does not meet the assumption for parametric testing, a Kruskal–Wallis test for more than two group comparisons (post hoc Wilcoxon rank sum test, p-value adjustment Bonferroni correction), or for a two group comparison, a Mann–Whitney U-test for independent samples was calculated. Significance levels against the respective controls were not significant (n.s) if $p>0.05$, or significant $*p<0.05$. Individual figure legends contain detailed information regarding the number of replicates, sample size and the applied statistical tests. Graphic illustrations were generated with *Biorender.com*.

## Acknowledgements

We thank Isabell Moskal, Gina Tillmann and Helene Schamber for the excellent technical support. This work was supported by

German Federal Ministry of Education and Research (BMBF) for "A systems-medicine approach towards distinct and shared resilience and pathological mechanisms of substance use disorders" (01ZX01909) and the Hector Stiftung II. We thank Valery Grinevich (Department of Neuropeptide Research, CIMH) for providing the AAV. This work was previously published as an abstract and poster at the following conferences: ISSCR 2021, eMed 2021 + 2022, GSCN 2022 + 2025, ICPR 2024.

## Additional information

### Competing interests

The authors declare that no competing interests exist.

### Funding

| Funder | Grant reference number | Author |
| --- | --- | --- |
| Hector Stiftung II | | Philipp Koch<br>Julia Ladewig |
| German Science Foundation (DFG) | TRR 265 | Rainer Spanagel<br>Marcus W Meinhardt |

| Funder | Grant reference number | Author |
|---|---|---|
| German Federal Ministry of Education and Research (BMBF) | 01ZX01909 | Rainer Spanagel<br>Philipp Koch<br>Julia Ladewig |

The funders had no role in study design, data collection and interpretation, or the decision to submit the work for publication.

## Author contributions

Malin Schmidt, Conceptualization, Data curation, Formal analysis, Validation, Investigation, Visualization, Methodology, Writing – original draft, Project administration, Writing – review and editing; Anne Hoffrichter, Data curation, Formal analysis, Validation, Methodology, Writing – review and editing; Mahnaz Davoudi, Data curation, Formal analysis, Validation, Investigation, Methodology; Sandra Horschitz, Marcus W Meinhardt, Resources, Writing – review and editing; Thorsten Lau, Julia Ladewig, Supervision, Writing – review and editing; Rainer Spanagel, Resources, Supervision, Writing – review and editing; Georg Köhr, Data curation, Formal analysis, Supervision, Investigation, Methodology, Writing – review and editing; Philipp Koch, Conceptualization, Resources, Supervision, Funding acquisition, Project administration, Writing – review and editing

## Author ORCIDs

Malin Schmidt ⓘ https://orcid.org/0009-0007-4907-8908
Anne Hoffrichter ⓘ https://orcid.org/0000-0001-6009-7826
Mahnaz Davoudi ⓘ https://orcid.org/0000-0001-9806-3496
Sandra Horschitz ⓘ https://orcid.org/0000-0001-9295-5493
Thorsten Lau ⓘ https://orcid.org/0009-0009-4390-3465
Marcus W Meinhardt ⓘ https://orcid.org/0000-0002-5103-0731
Rainer Spanagel ⓘ https://orcid.org/0000-0003-2151-4521
Julia Ladewig ⓘ https://orcid.org/0000-0002-5943-7990
Philipp Koch ⓘ https://orcid.org/0000-0003-3713-8786

Reviewer #1 (Public review): https://doi.org/10.7554/eLife.104006.3.sa1
Reviewer #2 (Public review): https://doi.org/10.7554/eLife.104006.3.sa2
Author response https://doi.org/10.7554/eLife.104006.3.sa3

# Additional files

## Supplementary files

MDAR checklist

Supplementary file 1. List of primers and antibodies used.

## Data availability

The R code used to analyze RNAseq data is available at https://github.com/ahoffrichter/Schmidt_et_al_2025 (copy archived at *Hoffrichter, 2026*). All data associated with the article is available on Dryad (https://doi.org/10.5061/dryad.xsj3tx9w3).

The following dataset was generated:

| Author(s) | Year | Dataset title | Dataset URL | Database and Identifier |
|---|---|---|---|---|
| Schmidt M, Hoffrichter A, Davoudi M, Horschitz S, Lau T, Meinhardt MW, Spanagel R, Ladewig J, Köhr G, Koch P | 2026 | Psilocin fosters neuroplasticity in iPSC-derived human cortical neurons | https://doi.org/10.5061/dryad.xsj3tx9w3 | Dryad Digital Repository, 10.5061/dryad.xsj3tx9w3 |

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
