## [Editor Report · eLife Assessment]

This **fundamental** study reports the effects of the psychedelic drug psilocin on iPSC-derived human cortical neurons, analyzing different aspects of structural and functional neuronal plasticity. The evidence is **convincing** and supports the value of using iPSC-derived human cortical neurons for testing the potentially translational effects of psilocin and other psychedelic-related compounds.

---

## [Referee Report · Reviewer #1 (Public review)]

Summary:

This study reports the effects of psilocin on iPSC-derived human cortical neurons.

Strengths:

The characterization was comprehensive, involving immunohistochemistry of various markers, 5-HT2A receptors, BDNF, and TrkB, transcriptomics analyses, morphological determination, electrophysiology, and finally synaptic protein measurements. The results are in close agreement with prior work (PMID 29898390) on rat cultured cortical neurons. Nevertheless, there is value in confirming those earlier findings and furthermore to demonstrate the effects in human neurons, which are important for translation. The genetic, proteomics, and cell structure analyses used in this paper are its major strength. The study supports the value of using iPSC-derived human cortical neurons for drug development involving psychedelics-related compounds.

Weaknesses:

(1) Line 140: 5-HT2A receptor expression was found via immunocytochemistry to reside in the somatodendritic and axonal compartments. However, prior work from ex vivo tissue using electron microscopy has found predominantly 5-HT2A receptor expression in the somatodendritic compartment (PMID: 12535944). Was this antibody validated to be 5-HT2A receptor-specific? Can the authors reason why the discrepancy may arise, and if the axonal expression is specific to the cultured neurons?

(2) Line 143: It would be helpful to specify the dose of psilocin tested, and describe how this dose was chosen.

(3) Figure 1: The interpretation is that the differential internalization in the axonal and somatodendritic compartments is time-dependent. However, given that only one dose is tested, it is also possible that this reflects dose dependence, with the longer time exposure leading to higher dose exposure, so these variables are related. That is, if a higher dose is given, internalization may also be observed after 10 minutes in the dendritic compartment.

(4) Figure 3 & 4: What is the 'control' here? A more appropriate control for the 24 hours after psilocin application would be 24 hours after vehicle application. Here the authors are looking at before and after, but the factor of time elapsed and perturbation via application is not controlled for.

(5) The sample size was not clearly described. In the figure legend, N = the number of neurites is provided, but it is unclear how many cells have been analyzed, and then how many of those cells belong to the same culture. These are important sample size information that should be provided. Relatedly, statistical analyses should consider that the neurites from the same cells are not independent. If the neurites indeed come from the same cells, then the sample size is much smaller and a statistical analysis considering the nested nature of the data should be used.

Comments on revisions:

The authors performed substantial experiments to check validity of the HTR2A antibody for the revision. Briefly, they found that western blot shows a single band, abolished by a blocking peptide, in neural progenitors and iPSC-derived neurons, suggesting positive results. However, they also detected immunofluorescence signals in HEK293 and HeLa cells, which do not express 5-HT2A receptors as scRNAseq analysis of these cells show complete absence of the transcript. Therefore the antibody has epitope-selective binding but also has some non-specific binding, precluding its use. The authors rightfully removed the data related to the antibody in the revised manuscript. The account is repeated here to highlight to anyone who may find the information helpful. Overall, the additional results added rigor to the study.

---

## [Referee Report · Reviewer #2 (Public review)]

In this article, Schmidt et al use iPSC-derived human cortical neurons to test the effects the psychedelic psilocin in different models of neuroplasticity.

Using human iPSC-derived cortical neurons, the authors test the expression of 5-HT2A and subcellular distribution, as well as the effect of different times of exposure to psilocin on 5-HT2A expression. The authors evaluated the effect of the 5-HT2 antagonist ketanserin, as well as the inhibition of dynamin-dependent endocytic pathways with dynasore. Gene expression and plasticity (structural and functional) was also evaluated after different times of exposure to psilocin.

In general, results are interesting since they use the iPSC to evaluate the potentially translationally relevant effects of psilocin (the active metabolite of the psychedelic psilocybin).

Comments on revisions:

The authors have addressed all of my previous concerns. A particular strength of the rebuttal is that the authors corroborated the lack of selectivity/specificity of the anti-5-HT2A antibody used in earlier versions of the manuscript.

---

## [Author Response]

The following is the authors’ response to the original reviews.

**Reviewer #1:**
Comment 1: 5-HT2A Antibody SpecificityWas this antibody validated to be 5-HT2A receptor-specific? Can the authors reason why the discrepancy may arise, and if the axonal expression is specific to the cultured neurons?

We performed extensive validation of the anti-5-HT2A receptor antibody (Alomone #ASR-033), which is summarized in the accompanying Author response images:

Positive findings (Author response image 1c-e, Author response image 2a): (1) Western blot showed a single band at the expected molecular weight (~50 kDa) in neural progenitors and iPSCderived neurons. (2) The blocking peptide (#BLP-SR033) abolished Western blot bands and markedly reduced immunofluorescence signals in neurons, confirming epitope-specific binding.

Negative findings (Author response image 1a-b, Author response image 2a-b, Author response image 3): (1) We detected positive immunofluorescence signals in HEK293 and HeLa cells (Author response image 1a-b), which do not express 5-HT2AR. (2) Western blot also showed bands in HEK293 and HeLa cells (Author response image 2a-b). (3) Single-cell RNA-seq analysis of HEK293T cells confirmed complete absence of *HTR2A* expression (Author response image 3a). (4) qPCR showed no detectable *HTR2A* transcripts in iPSCs or HeLa cells (Ct > 36), while neural progenitors and neurons showed clear expression (Author response image 3b). (5) siRNA knockdown experiments failed to produce a corresponding decrease in immunofluorescence or Western blot signals, despite reduced *HTR2A* transcript levels (data not shown).

BLAST analysis: Protein BLAST analysis of the 13-amino acid immunogenic peptide sequence identified the human 5-HT2A receptor as the top hit (9/13 amino acids overlap). However, shorter sequence similarities were also found with other proteins, including APPBP1 (6/9 amino acids), Immunoglobulin Heavy Chain (6/7 amino acids), and Interleukin31 receptor (6/8 amino acids). While these partial homologies do not provide a definitive mechanistic explanation for the observed off-target binding, they illustrate that the epitope sequence is not entirely unique to the 5-HT2A receptor.

Conclusion: While our validation confirmed epitope-specific binding (blocking peptide effective in neurons), the antibody clearly detects something in cells that demonstrably lack HTR2A gene expression. This indicates off-target binding to other proteins sharing the epitope sequence. We have therefore removed all antibody-based 5-HT2A receptor experiments from the revised manuscript. This includes the receptor internalization data from Figure 1. The remaining findings (BDNF upregulation, gene expression changes, morphological effects, electrophysiology) are supported by independent methods including pharmacological blockade with ketanserin.

Comment 2: Psilocin Dose SelectionIt would be helpful to specify the dose of psilocin tested, and describe how this dose was chosen.

We used 10 µM psilocin based on: (1) The seminal study by Ly et al. (2018), which demonstrated neuroplasticity effects at this concentration in rat cortical neurons. (2) Our own dose-response experiments (Figure S2B) showing maximal BDNF increase at 10 µM compared to lower concentrations (10 nM, 100 nM, 1 µM). We have clarified this in the revised Methods section.

Comment 3: Dose vs. Time DependenceGiven that only one dose is tested, it is also possible that this reflects dose dependence, with the longer time exposure leading to higher dose exposure.

We agree that dose dependence cannot be excluded with our current experimental design. This point is now moot as we have removed the 5-HT2A receptor internalization experiments from the manuscript. Future studies in our group will address dose-dependent effects on other readouts.

Comment 4: Control ConditionsWhat is the 'control' here? A more appropriate control would be 24 hours after vehicle application.

The control condition is indeed a vehicle (DMSO) control collected at the same time point as the experimental condition (i.e., 24 hrs post-treatment). We have clarified this in the revised figure legends and Methods section to avoid confusion.

Comment 5: Sample Size DescriptionThe sample size was not clearly described. Statistical analyses should consider that neurites from the same cells are not independent.

We have expanded the sample size descriptions in the figure legends. Analyses were performed using 5-10 microscope images per condition, with 15 ROIs per image, across at least two independent differentiations from two genetic backgrounds. Regarding independence: each neurite segment exists within a distinct microenvironment and can be considered an independent measurement unit, consistent with established practices in the field (Paul et al., 2021, CNS Neurosci Ther). We acknowledge this increases statistical power and have noted this in the Methods.

**Reviewer #2:**
Comment 1: 5-HT2A Antibody ValidationWithout validation (using for example knockdown techniques to decrease expression of 5HT2A), the experiments using this antibody should be excluded from the manuscript.

We agree with this assessment. As detailed in our response to Reviewer 1 (Comment 1) and documented in the Response to Reviewer Figure, our extensive validation attempts—including siRNA knockdown—could not conclusively demonstrate antibody specificity. We have removed all antibody-based 5-HT2A receptor experiments from the revised manuscript.

Comment 2: Serotonin in Cell MediaDid the authors evaluate whether 5-HT is present in the cell media?

The cell culture media used in our experiments does not contain serotonin. We have explicitly stated this in the revised Methods section.

Comment 3: Statistical Analysis of Figure S1FSome of the datasets are not statistically analyzed, such as Figure S1F.

Figure S1F related to the 5-HT2A receptor experiments and has been removed from the revised manuscript along with the associated data.

Comment 4: Translational Validity of Prolonged ExposureThe authors continuously exposed cells to psilocin for hours or days. Since this is not the model of what occurs in vivo, the findings lack translational validity.

We acknowledge this limitation. Most experiments (BDNF, gene expression, branching) were conducted 24–48 hrs after a brief 10-minute exposure, which better reflects the *in vivo* situation. Prolonged exposures (96 hrs) were used specifically for synaptogenesis experiments based on literature showing that repeated LSD administration enhances spine density (Inserra et al., 2022; De Gregorio et al., 2022). Our *in vitro* system lacks metabolizing enzymes and glial cells, which may introduce temporal biases. We have added a discussion of these limitations in the revised manuscript.

Comment 5: Ketanserin Effect on BDNFIn Figure 2E, ketanserin by itself seems to reduce BDNF density. How do the authors conclude that ketanserin blocks psi-induced effects?

We identified that one cell line (Ctrl 1) with inherently higher BDNF density was inadvertently excluded from the ketanserin-only condition. After removing Ctrl 1 from all conditions and reanalyzing, the difference between Ctrl and Ket alone is no longer significant. The significant difference between Psi+Ket and Ket alone demonstrate that psilocin exerts effects that ketanserin can block, consistent with 5-HT2A receptor mediation. The revised figure and statistical analysis are included in the updated manuscript.

Comment 6: mCherry Localization mCherry (Fig 4A) seems to be retained in the nucleus.

The CamKII promoter drives expression of cytoplasmic mCherry, which fills the entire neuron including soma, dendrites, and axons. The apparent nuclear signal reflects mCherry accumulation in the soma, which surrounds the nucleus. The images clearly show mCherry extending into neurites, which was essential for our Sholl analysis of neuronal complexity.

Comment 7: Reference 36Reference 36 is a review article that does not mention psilocin.

Our statement refers broadly to serotonergic psychedelics increasing neurotrophic factors. Reference 36 (Colaço et al., 2020) examines ayahuasca, which contains the serotonergic psychedelic DMT. We have revised the text to clarify this point.

Summary of Major Revisions

(1) Removed all 5-HT2A receptor antibody-based experiments from Figure 1 and supplementary figures due to inconclusive specificity validation. An Author response image documenting our validation attempts is provided.

(2) Clarified control conditions (vehicle controls at matched time points) in figure legends.

(3) Expanded sample size descriptions in Methods and figure legends.

(4) Re-analyzed ketanserin experiments with consistent cell line inclusion.

(5) Added discussion of translational limitations.

(6) Added new Figure S5 summarizing proposed signaling pathways.

(7) Expanded discussion on the relevance of iPSC-derived neurons for drug development.

**Author response image 1. sa3fig1:** Immunostaining for 5-HT2A receptor across cell types and peptide-blocking control. (a) HEK293 cells display a positive immunofluorescent signal despite not endogenously expressing 5-HT2AR, indicating nonspecific antibody reactivity. (b) HeLa cells also exhibit a positive signal despite lacking endogenous 5-HT2AR expression, further demonstrating nonspecific antibody binding in non-expressing cell types. (c) Neural progenitor cells show clear positive 5-HT2AR staining. (d) iPSC-derived neurons exhibit robust and well-defined 5-HT2AR staining. (e) Application of the Alomone 5-HT2AR blocking peptide (#BLP-SR033) markedly reduces neuronal signal intensity, supporting epitope-specific binding.

**Author response image 2. sa3fig2:** Western blot analysis of 5-HT2A receptor abundance and peptide-blocking control. (a-b) In line with the immunofluorescence a single band is detected in iPSCs, HEK cells, neural progenitors, iPSC-derived neurons and (b) HeLa cells. (a) Preincubation of the primary antibody with the corresponding blocking peptide abolishes this band across all samples, consistent with specific binding of the antibody to its intended epitope.

**Author response image 3. sa3fig3:** Lack of detectable 5-HT2AR expression in HEK and HeLa cells. (a) Analysis of a human-only HEK293T single-cell RNA-seq dataset (10x Genomics; here, accessed 2025-11-25) shows no meaningful *HTR2A* expression, whereas other genes such as *GAPDH*, *TP53*, *MYC*, and *ACTB* are robustly detected. Consistently, evaluation of a “Barnyard” dataset - an equal mixture of human HEK293T and mouse NIH3T3 cells (10x Genomics; here, accessed 2025-1125) reveals only ~4 of ~10,000 droplets with minimal HTR2A signal, confirming the absence of meaningful expression.(b) (b) qPCR analysis further demonstrates no detectable *HTR2A* transcripts in iPSCs or HeLa cells (Ct > 36), while neural progenitors and iPSC-derived cortical neurons show expression when normalized to housekeeping genes *GAPDH* and *TBP*.